# Photo-thermo semi-hydrogenation of acetylene on Pd$_1$/TiO$_2$ single-atom catalyst

Yalin Guo[1,2,7], Yike Huang [1,2,7], Bin Zeng[2,3], Bing Han[1,2], Mohcin AKRI[1], Ming Shi[2,3], Yue Zhao[3], Qinghe Li[1], Yang Su[1], Lin Li[1], Qike Jiang[4], Yi-Tao Cui[5], Lei Li[6], Rengui Li [3✉], Botao Qiao [1✉] & Tao Zhang [1]

Semi-hydrogenation of acetylene in excess ethylene is a key industrial process for ethylene purification. Supported Pd catalysts have attracted most attention due to their superior intrinsic activity but often suffer from low selectivity. Pd single-atom catalysts (SACs) are promising to significantly improve the selectivity, but the activity needs to be improved and the feasible preparation of Pd SACs remains a grand challenge. Here, we report a simple strategy to construct Pd$_1$/TiO$_2$ SACs by selectively encapsulating the co-existed small amount of Pd nanoclusters/nanoparticles based on their different strong metal-support interaction (SMSI) occurrence conditions. In addition, photo-thermo catalysis has been applied to this process where a much-improved catalytic activity was obtained. Detailed characterization combined with DFT calculation suggests that photo-induced electrons transferred from TiO$_2$ to the adjacent Pd atoms facilitate the activation of acetylene. This work offers an opportunity to develop highly stable Pd SACs for efficient catalytic semi-hydrogenation process.

---

[1] CAS Key Laboratory of Science and Technology on Applied Catalysis, Dalian Institute of Chemical Physics, Chinese Academy of Sciences, Dalian, China. [2] University of Chinese Academy of Sciences, Beijing, China. [3] State Key Laboratory of Catalysis, Dalian National Laboratory for Clean Energy, Dalian Institute of Chemical Physics, Chinese Academy of Sciences, Dalian, China. [4] Dalian National Laboratory for Clean Energy, Dalian Institute of Chemical Physics, Chinese Academy of Sciences, Dalian, China. [5] SANKA High Technology Co. Ltd. 90-1, Tatsuno, Hyogo, Japan. [6] Synchrotron Radiation Research Center, Hyogo Science and Technology Association, Hyogo, Japan. [7] These authors contributed equally: Yalin Guo, Yike Huang. ✉email: rgli@dicp.ac.cn; bqiao@dicp.ac.cn

Ethylene, one of the basic building blocks to produce plastic and key chemicals, is predominantly manufactured from steam cracking of hydrocarbons thus usually concomitant with small amount of co-produced acetylene. The co-existed acetylene can severely poison the downstream catalyst for ethylene transformation, thereby has to be diminished to an acceptable level (often <5 ppm)[1,2]. Diverse methods have been developed to eliminate the acetylene impurity among which the electrocatalysis has been proven a green chemistry approach[3,4]. For example, Shi et al. reported a room-temperature electrochemical reduction strategy of acetylene over a layered double hydroxide (LDH)-derived Cu catalyst, which manifested high catalytic performance but suffers from unaddressed issues for large-scale applications, such as the low cell energy efficiency[3]. On the other hand, thermocatalytic semi-hydrogenation of acetylene into ethylene seems more efficient, and has been extensively applied in industry for decades. Among various catalysts explored, supported Pd catalysts have attracted most attention on account of their superior intrinsic activity. Unfortunately, the low selectivity, especially at the full conversion of acetylene, has long been a serious concern. Several strategies based on the "active site isolation" concept, such as selective poisoning/covering special Pd sites (Lindlar catalysts[5]) or forming Pd-M alloy/intermetallic compounds (industrially used Ag-Pd/Al$_2$O$_3$ catalysts)[6–11] to weaken the adsorption of ethylene have been frequently used to improve the selectivity[12–21], which are, nevertheless, often at the cost of activity loss due to the presence of substantial inaccessible Pd sites.

Single-atom catalysts (SACs) have attracted rapid growing interests as a new frontier in heterogeneous catalysis field[22,23]. In SACs, isolated metal atoms are spatially separated and uniformly distributed on the surface of the support, perfectly meeting the "active-site isolation" concept while simultaneously maximizing the metal utilization efficiency. Hence, SACs have been regarded as an ideal candidate for semi-hydrogenation of alkyne and have shown promising catalytic performance[24–29]. Unfortunately, SACs are generally less effective for H$_2$ activation, giving rise to a depressed hydrogenation activity[30]. Moreover, to maintain the isolated dispersion and good stability of SACs, a very low metal loading is often used. This is particularly true for Pd-based SACs[31]. All these render the semi-hydrogenation of acetylene on Pd SACs working currently at elevated temperatures. Therefore, to meet industrial application, it is necessary to develop stable and efficient Pd SACs meanwhile lowering the working temperature.

Strong metal–support interaction (SMSI), a topic being extensively studied for more than 40 years in heterogeneous catalysis area[32,33], has sparked renewed interests due to their potential in modifying catalyst performance, and especially in stabilizing catalysts[34–40]. Recently, we found that isolated Pt atoms supported on TiO$_2$ can manifest classical SMSI[34] but at a much higher reduction temperature. A feature of this finding is that the co-existed nanoparticles (NPs) can be selectively encapsulated while single atoms keep exposed through reduction at suitable temperatures. This finding might be extended to TiO$_2$ supported other metal catalysts thus providing a new strategy to construct stable SACs. On the other hand, Photo-thermo catalysis is an emerging sub-discipline that involves the integration of thermo- and photocatalytic processes, which is distinct from the traditional thermo-catalysis because photogenerated carriers can directly transfer into the orbitals of adsorbed molecules to promote their desorption, dissociation, or activation thus trigger the chemical reaction, giving rise to a totally different reaction pathway[41–44]. Recent pioneering studies have demonstrated that the coupling of thermo- and photocatalytic processes overcomes the low activity in photocatalysis and high reaction barrier in thermocatalysis, thus offering a promising strategy to promote the activity and/or selectivity for various meaningful reactions,

such as hydrogenation, oxidation, CO$_2$ reduction, Fischer-Tropsch synthesis, water–gas shift reaction[45–52]. Despite of these great progress, whether photo-thermocatalysis is possible to boost semi-hydrogenation of acetylene still remains inconclusive. Few studies related to photocatalysis of selective hydrogenation for nitrobenzene[53], benzaldehyde[54], and alkynyl group[55–59] were reported, but the related works for photo-thermocatalytic acetylene semi-hydrogenation are limited. Swearer et al. firstly used Pd NPs and aluminum nanocrystals (AlNC) to construct a heterometallic antenna-reactor complexes photocatalyst for semi-hydrogenation of acetylene but with a low product yield[60]. The other one is working at relatively high temperature by converting photo into heat rather than an integration of photo-thermo catalysis process at lower temperature[28].

Herein, we report a simple yet general strategy to improve the selectivity of TiO$_2$ supported Pd catalysts prepared by a variety of methods via selectively encapsulating the co-existed small amount of Pd nanoclusters/nanoparticles (NPs) due to their different SMSI occurrence temperatures. In addition, on account of the superior photocatalysis of TiO$_2$ support, a much-improved catalytic activity was obtained by integrating photo-thermo catalysis and a dramatically decreased working temperature of as low as 70 °C was realized. Detailed studies reveal that photo-induced electrons transferred from TiO$_2$ to the adjacent Pd atoms facilitate the activation of acetylene and thus benefit the photo-thermo catalytic semi-hydrogenation reaction.

## Results

**Synthesis and structural characterization of Pd/TiO$_2$.** The Pd/TiO$_2$ catalyst was firstly synthesized by ball milling based on the so called "precursor-dilution" strategy[61–64] to obtain better dispersion but we will propose later that much more practical methods such as strong electrostatic adsorption (SEA) and even impregnation methods also work well. The obtained Pd/TiO$_2$ was reduced at 200 °C and 600 °C, denoted as Pd/TiO$_2$-200H, and Pd/TiO$_2$-600H, respectively. For comparison, the Pd/TiO$_2$-600H catalyst was re-oxidized by 10 vol% O$_2$ at 300 °C, denoted as Pd/TiO$_2$-600H-O300. In addition, pure rutile supported catalysts were also prepared and tested in similar procedures.

The BET specific surface area was measured to be about 70 m$^2$ g$^{-1}$ by N$_2$ physical adsorption–desorption process, and the incorporation of Pd did not change the surface area much, Supplementary Fig. 1. As shown in Supplementary Fig. 2, the X-ray diffraction (XRD) spectrum of the synthesized TiO$_2$ support displays typical patterns of both anatase and rutile, suggesting a mixture structure. After loading of Pd, and even after reduction at different temperatures and re-oxidation, there is no obvious structure change of the TiO$_2$ support as evidenced by the similar diffraction patterns of various catalysts to that of TiO$_2$ support. In addition, no any diffraction pattern associated with Pd species is observed, suggesting either the Pd is highly dispersed or the Pd loading is too low to be detected. The high dispersion of Pd was further examined by aberration-corrected scanning transmission electron microscopy (AC-STEM). High-magnification high-angle annular dark-field (HAADF) STEM images reveal the presence of relatively high density of Pd single atoms on all catalysts, Fig. 1a–d. Meanwhile some other HAADF-STEM images indicate the presence of small portion of Pd NPs, Supplementary Fig. 3. It stressed the great difficulty in fabricating "absolute" Pd SACs (presence of only isolated single atoms without any clusters/NPs) even with such an effective ball-milling method[61–64]. To our knowledge, so far the Pd SACs with relatively high metal loading on non-carbon supports have been rarely reported[65–67].

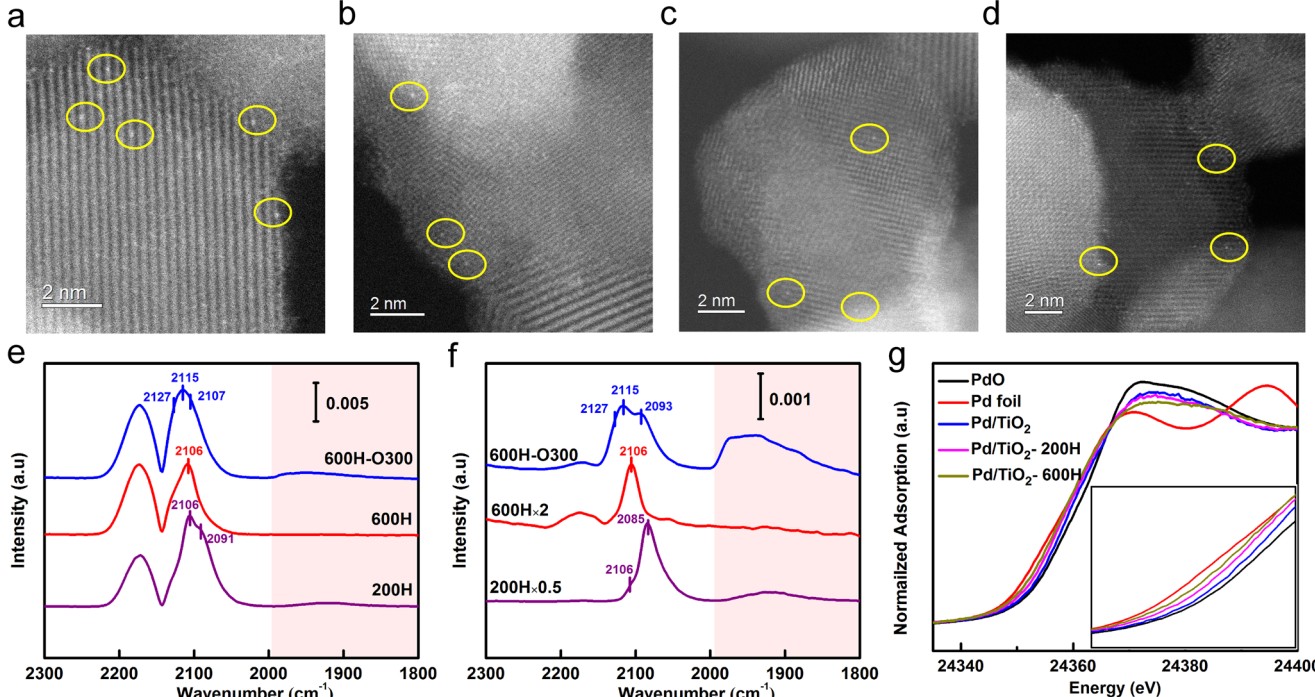

**Fig. 1 Structural characterization of Pd/TiO$_2$ serial catalysts. a–d** AC-HAADF-STEM images of **a** Pd/TiO$_2$, **b** Pd/TiO$_2$-200H, **c** Pd/TiO$_2$-600H, and **d** Pd/TiO$_2$-600H-O300; Pd single atoms are highlighted in yellow circles. **e, f** DRIFT spectra of CO adsorption on Pd/TiO$_2$-200H, Pd/TiO$_2$-600H, and Pd/TiO$_2$-600H-O300 **e** at CO saturation adsorption and **f** upon He purging for 2.5 min at room temperature. **g** XANES spectra of different Pt/TiO$_2$ catalysts at Pd K-edge absorption edge.

Very recently, we discovered that Pt single atoms on TiO$_2$ can manifest classical SMSI upon reduction but at a much higher reduction temperature compared with Pt NPs. The most meaningful feature of this discovery is that the NP active sites can be selectively encapsulated upon reduction at certain temperatures, therefore the catalytic performance can be finely tuned[34–40]. We believe this scenario is general and may be extended to TiO$_2$ supported Pd catalysts to distinctly refine their catalytic performance and we will prove this in the following.

Diffuse reflectance infrared Fourier transform (DRIFT) spectra of CO adsorption were first employed to study the SMSI state of our Pd/TiO$_2$ sample and the results are presented in Fig. 1e, f and Supplementary Fig. 4. For CO saturation adsorption on Pd/TiO$_2$-200H, Fig. 1e, two peaks centered at 2106 and 2091 cm$^{-1}$ and a broad band existed in the range of 1840–1990 cm$^{-1}$ were observed in addition to the two gas phase CO bands. The former two are ascribed to linear CO adsorption on Pd single atoms and Pd NPs[68], respectively, while the latter band is ascribed to the bridged and/or three-hollowed CO adsorption on Pd NPs. The linear CO adsorption on Pd single atoms (2106 cm$^{-1}$) can be further verified by the weaker adsorption (i.e., faster desorption upon He purge) and no frequency shift with CO coverage, which is in contrast to a stronger CO adsorption and a frequency shift with CO coverage change on Pd NPs (from 2091 to 2082 cm$^{-1}$), Fig. 1f and Supplementary Fig. 4. The DRIFT characterization indicated the co-presence of Pd single atoms and Pd NPs, in consistent with the AC-STEM characterization results. After reduction at 600 °C, both linear and bridged CO adsorption on Pd NPs disappeared completely, Fig. 1e, suggesting the Pd NPs were encapsulated by reduced TiO$_x$ layer, a typical characteristic of SMSI. On the contrary, CO adsorption on Pd single atoms remains almost unchanged, suggesting the occurrence of SMSI on Pd atoms is harder than that on Pd NPs. The encapsulation of Pd NPs was further examined by AC-STEM, Fig. 2. The encapsulation layer can be clearly observed in bright field (BF) image, Fig. 2a, and the

TiO$_x$ nature was confirmed by electron energy loss spectroscopy (EELS), Fig. 2b. After calcination at 300 °C, all CO adsorption on Pd NPs appears again, suggesting the encapsulation is reversible, another typical characteristic of SMSI. The retreated cover layer was revealed by both BF-STEM image and EELS spectra, Fig. 2c, d. Significantly, the still co-existence of single Pd atoms and Pd NPs illustrates that no aggregation happened during the reversible encapsulation process as shown in Fig. 1 and Supplementary Fig. 3. This set of characterization unambiguously indicates that the co-presented small amount of Pd NPs can be selectively encapsulated by TiO$_x$ layer upon reduction at 600 °C.

The electronic properties of Pd species with different treatments were studied by X-ray absorption near-edge structure (XANES) spectroscopy with Pd foil and PdO as reference samples. As shown in Fig. 1g, XANES spectra display that the Pd K-edge absorption edge for Pd/TiO$_2$, Pd/TiO$_2$-200H, and Pd/TiO$_2$-600H all located between that of Pd foil and PdO, indicating the existence of positively charged Pd species. Additionally, the spectrum of Pd/TiO$_2$-600H is closer to that of Pd foil, suggesting a lower chemical state with deeper reduction. The positive chemical state was contributed by Pd single atoms. Since the fact that under SMSI state Pd NPs often existed as metallic or even negatively charged state[69], the slightly positive chemical state of Pd suggests that the proportion of Pd NPs in the catalyst is very low. The valence state of Pd species on Pd/TiO$_2$-600H was further examined by X-ray photoelectron spectroscopy (XPS), Supplementary Fig. 5. It shows a mixture of Pd$^0$ and Pd$^{\delta+}$, consistent well with the XANES result and further verifying the high valence of Pd single atoms.

**Catalytic performance evaluation.** Above results exhibited that the small amount of co-existed Pd clusters/NPs on TiO$_2$ supported Pd SACs can be selectively encapsulated meanwhile Pd single atoms were kept exposed thanks to their different SMSI

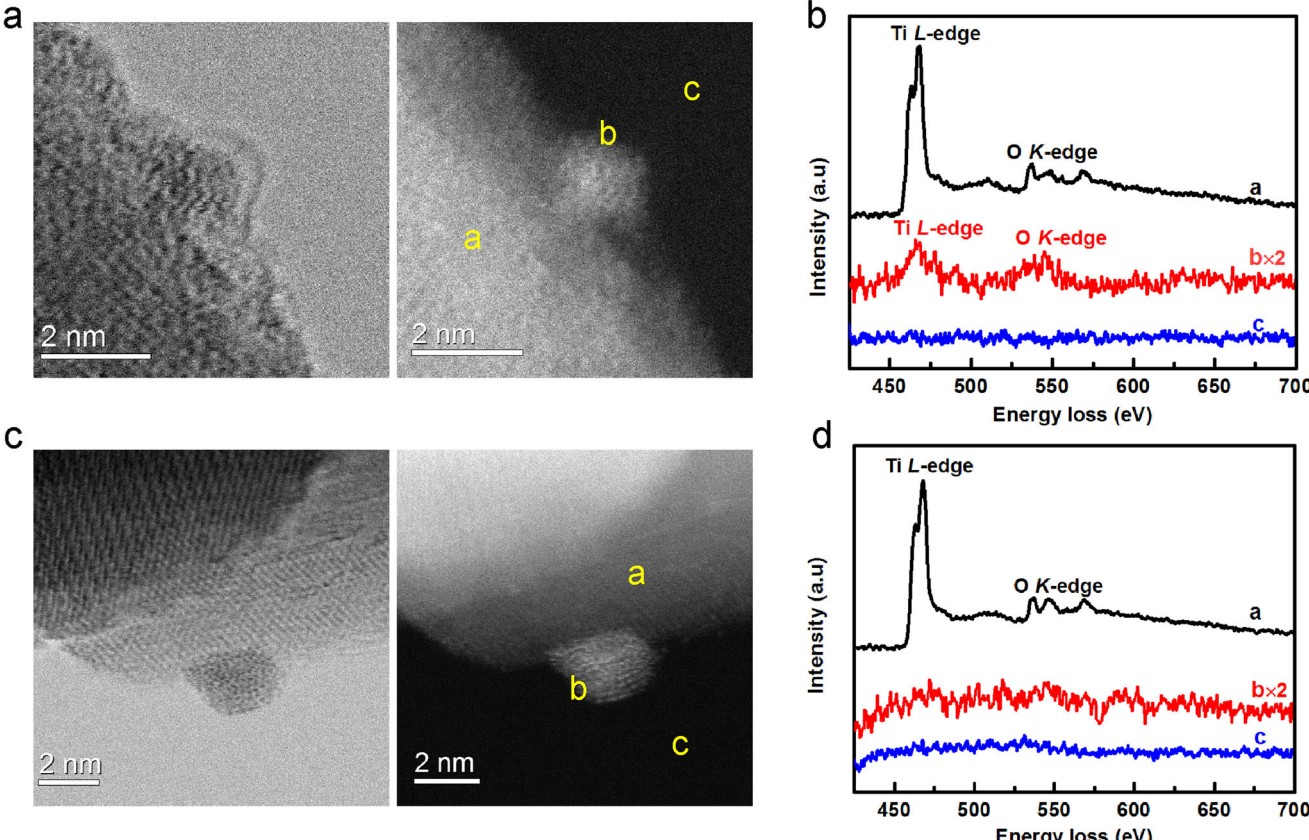

**Fig. 2 Structural characterization of Pd/TiO$_2$ serial catalysts. a** BF-STEM and HAADF-STEM images of Pd/TiO$_2$-600H. **b** EELS spectra of Pd/TiO$_2$-600H. **c** BF-STEM and HAADF-STEM images of Pd/TiO$_2$-600H-O300. **d** EELS spectra of Pd/TiO$_2$-600H-O300.

occurrence temperatures. This feature is highly valuable in manipulating the catalytic performance. We testified this in semi-hydrogenation of acetylene which is very sensitive to the isolation nature of the active sites. As shown in Fig. 3a, Pd/TiO$_2$-200H exhibited a high catalytic activity and the conversion of acetylene reached 100% at 120 °C at a high weight hourly space velocity (WHSV = 120,000 mL h$^{-1}$ g$_{cat}$$^{-1}$). However, the selectivity of ethylene is as low as −300% due to the over hydrogenation of acetylene and hydrogenation of ethylene raw materials, Fig. 3b. It further decreased to −500% with reaction temperature increase. Remarkably, Pd/TiO$_2$-600H exhibits a much-improved ethylene selectivity (from −500% to 40%) without any activity compromise in a wide reaction temperature window of 100−200 °C. In addition, in a 40 h long-term test at 120 °C, the selectivity only slightly decreased from 65 to 50% in the initial 30 h and kept almost unchanged thereafter, suggesting a good catalyst durability, Fig. 3c. HAADF-STEM images of the used catalyst did not reveal detectable sintering of Pd single atoms/NPs, Supplementary Fig. 6, in consistent well with the good catalyst durability.

Above results demonstrated the application of Pd/TiO$_2$ catalysts in selective hydrogenation reaction by controlling the SMSI. Since the catalyst was prepared by a high-energy ball-milling process and the phthalocyanine precursors are costly, the process is less feasible for practical application. To verify the universality of this strategy, we prepared 0.15 wt% Pd/TiO$_2$ catalysts by SEA[70] (Pd/TiO$_2$-SEA) and incipient wet-impregnation (Pd/TiO$_2$-IWI) methods, which are more simple and practically applicable. Same scenario was observed on both catalysts that the selectivity after reduction at a high temperature (650 °C) was much improved compared with those reduced at 200 °C, with only a little activity decrease on both catalysts

(Supplementary Figs. 7, 8). The stability is also very good; for example, Pd/TiO$_2$-SEA catalyst tested at 120 °C and 150 °C displays negligible selectivity change after 30 h, Supplementary Fig. 8. Additionally, a similar scenario happened on rutile supported Pd catalysts, which show that the ethylene selectivity after reduction at a higher temperature 700 °C is much improved compared with that reduced at 200 °C, Supplementary Fig. 9. A slightly higher reduction temperature of 700 °C was adopted because rutile is slightly harder to form SMSI[34,71]. This result suggests that the crystal phase of the support has limited influence on the catalytic performance of SACs, in contrast to that on Pd nanocatalysts[72,73].

**Photo-thermo catalysis of semi-hydrogenation of acetylene.** Above results show that we have developed a general strategy to manipulate the catalytic performance, i.e., significantly improving the selectivity of acetylene semi-hydrogenation. However, the dissociation of H$_2$ usually goes to a heterolytic pathway on SACs, overcoming a higher barrier than that on the NPs with homolytic dissociation[29]. Lacking cooperation from neighboring metal atoms and losing metallic property, may decrease the ability of single metal atoms to dissociate H$_2$, leading to a lower intrinsic activity for SACs. Recently, photo-thermo catalysis, i.e., integrating photo- and thermo-catalysis to boost catalytic performance at more mild reaction condition, has attracted growing attention[45]. However, little attention has been paid on the photo-thermo catalysis of acetylene semi-hydrogenation so far[28,60]. Here we verified that photo-thermo catalysis could be a promising way for semi-hydrogenation of acetylene. A continuous flow fixed-bed reactor, as illustrated in Supplementary Fig. 10, was used for the

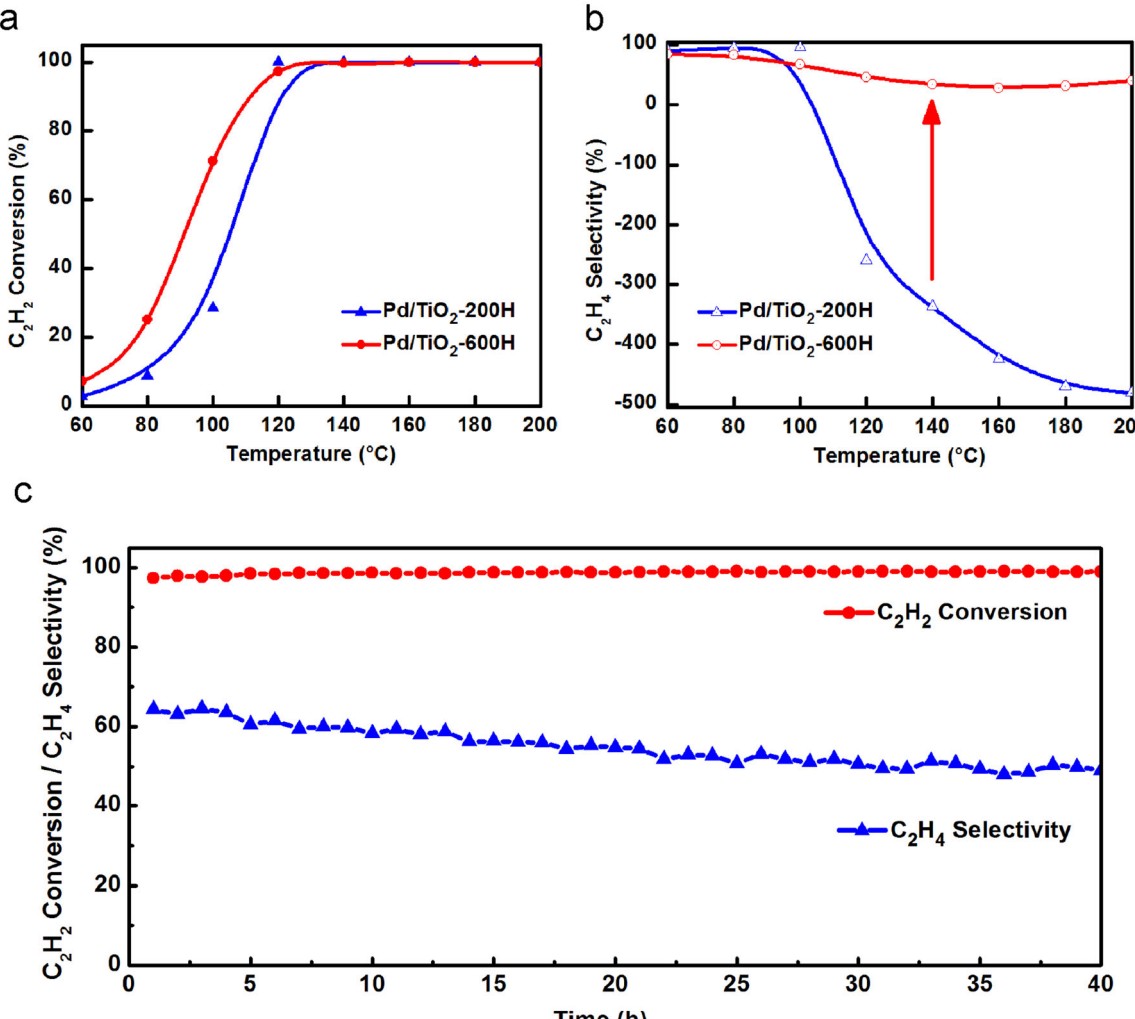

**Fig. 3 Catalytic performance of Pd/TiO$_2$ serial catalysts.** Acetylene conversion **a** and ethylene selectivity **b** as a function of temperature for acetylene semi-hydrogenation over Pd/TiO$_2$-200H and Pd/TiO$_2$-600H. **c** Durability test on Pd/TiO$_2$-600H at 120 °C for 40 h. Reaction conditions: 1 vol% C$_2$H$_2$, 10 vol% H$_2$, 20 vol% C$_2$H$_4$ balanced with He; WHSV = 120,000 mL h$^{-1}$ g$_{cat}$$^{-1}$.

photo-thermo catalytic semi-hydrogenation of acetylene. It turns out that the introduction of light irradiation can indeed boost the conversion of acetylene in semi-hydrogenation reaction remarkably (from 20 to 80%) with a decreased ethylene selectivity, Fig. 4 and Supplementary Fig. 11, indicating that the photogenerated charges may involve in the reaction process. The decreased selectivity upon irradiation might stem from the increased acetylene conversion as the ethylene selectivity is usually acetylene-conversion-dependent[1]. After an initial evaluation stage (first cycle), the activity increased significantly from 30 to 80% upon irradiation with an increased WHSV of 180,000 mL h$^{-1}$ g$_{cat}$$^{-1}$, then the conversion and selectivity maintain stable (see second cycle) which can be verified by tests with more cycles, Supplementary Fig. 12. The almost unchanged ethylene selectivity after second cycle further confirms the above conjecture that ethylene selectivity decrease is mainly related to the acetylene conversion, although influences from irradiation induced other changes, such as chemical and/or SMSI[74] state of Pd species, cannot be completely excluded.

To identify whether the improvement was induced by the thermal effect or not, we carefully measured the temperature under the light irradiation. As shown in Supplementary Fig. 10, the potential heat effect was minimized by using a water bath equipped with circulating water. It is suggested that the light can still

increase the reactor temperature by 10 °C below 100 °C. As a result, a 10 °C-correction was used when performing photo-thermo catalysis. It should be emphasized that upon light irradiation a very sharp response rather than a gradual increased process was observed, Fig. 4, demonstrating this is a photo-thermo catalysis process by integration of photo- and thermo-catalysis[45]. To further investigate the effect of light irradiation on semi-hydrogenation of acetylene, the wavelength-dependence experiment was conducted under different wavelength ranges (full spectrum, λ > 420 nm and λ > 480 nm). As shown in Fig. 4b, when the catalyst was irradiated only by visible light (λ > 420 nm), the increasement of acetylene conversion was significantly reduced compared with the full-spectrum case in Fig. 4a, which indicates that the photoexcitation of TiO$_2$ is critical in semi-hydrogenation of acetylene. Noted that a slight improvement is still present when the visible light irradiated, owing to the weak light absorption at the band edges of TiO$_2$. Further, a 480 nm cutoff filter was introduced for comparison, which can remove the possibility of the photoexcitation of TiO$_2$. As expected, no improvement can be seen under light condition (λ > 480 nm) compared with the dark case, Fig. 4c. Above results imply that the improvement in semi-hydrogenation of acetylene is significantly attributed to the photoexcitation of TiO$_2$, manifesting that the photogenerated charges involve and contribute to the semi-hydrogenation reaction. A control experiment using Al$_2$O$_3$ as

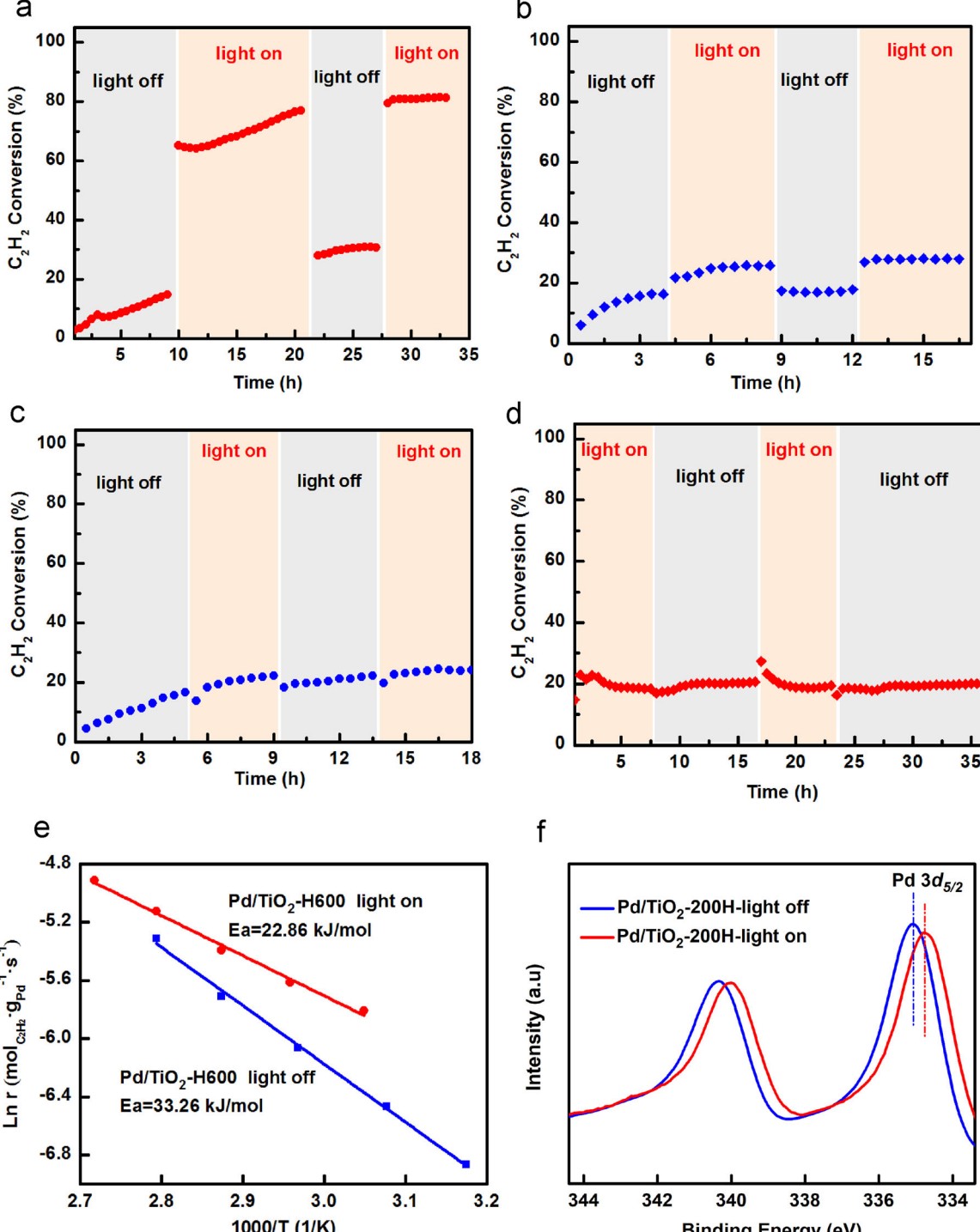

**Fig. 4 Catalytic performance of photo-thermo catalysis over Pd/TiO$_2$-600H, Pd/Al$_2$O$_3$ and XPS characterization. a–c** Acetylene conversion over the catalyst of 0.15 wt% Pd/TiO$_2$-600H at 70 °C in the dark and 60 °C upon **a** full-spectrum light irradiation, power density: 167 mW cm$^{-2}$, **b** visible light (λ > 420 nm), power density: 141 mW cm$^{-2}$ and **c** λ > 480 nm, power density: 127 mW cm$^{-2}$. Reaction conditions: 1 vol% C$_2$H$_2$, 10 vol% H$_2$, 20 vol% C$_2$H$_4$ balanced with He; WHSV = 180,000 mL h$^{-1}$ g$_{cat}$$^{-1}$. **d** Acetylene conversion over the catalyst of 0.036 wt% Pd/Al$_2$O$_3$ at 70 °C in the dark and 60 °C upon full-spectrum light irradiation with the similar reaction conditions. **e** Arrhenius plots for acetylene hydrogenation over the Pd/TiO$_2$-600H upon light irradiation or in the dark under the gas mixture of 1 vol% C$_2$H$_2$, 20 vol%H$_2$, 20 vol% C$_2$H$_4$ balanced with He. **f** Pd 3$d$ XPS spectra of Pd/TiO$_2$-200H after light irradiation by Xe lamp as well as the catalyst in dark.

a catalyst support to replace of TiO$_2$, which suggests no enhancements in conversion and selectivity were observed, Fig. 4d and Supplementary Fig. 11d, further verifying the superiority of photo-thermo catalysis on the Pd/TiO$_2$ catalyst. Additionally, an intensity-dependent experiment upon a full-spectrum light

irradiation was conducted at WHSV = 360,000 mL h$^{-1}$ g$_{cat}$$^{-1}$, Supplementary Fig. 13. The acetylene activity is boosted with the increased power density from 133 to 270 mW cm$^{-2}$ accompanied by an opposite trend in ethylene selectivity. Moreover, kinetic measurement suggests that the apparent activation energy in

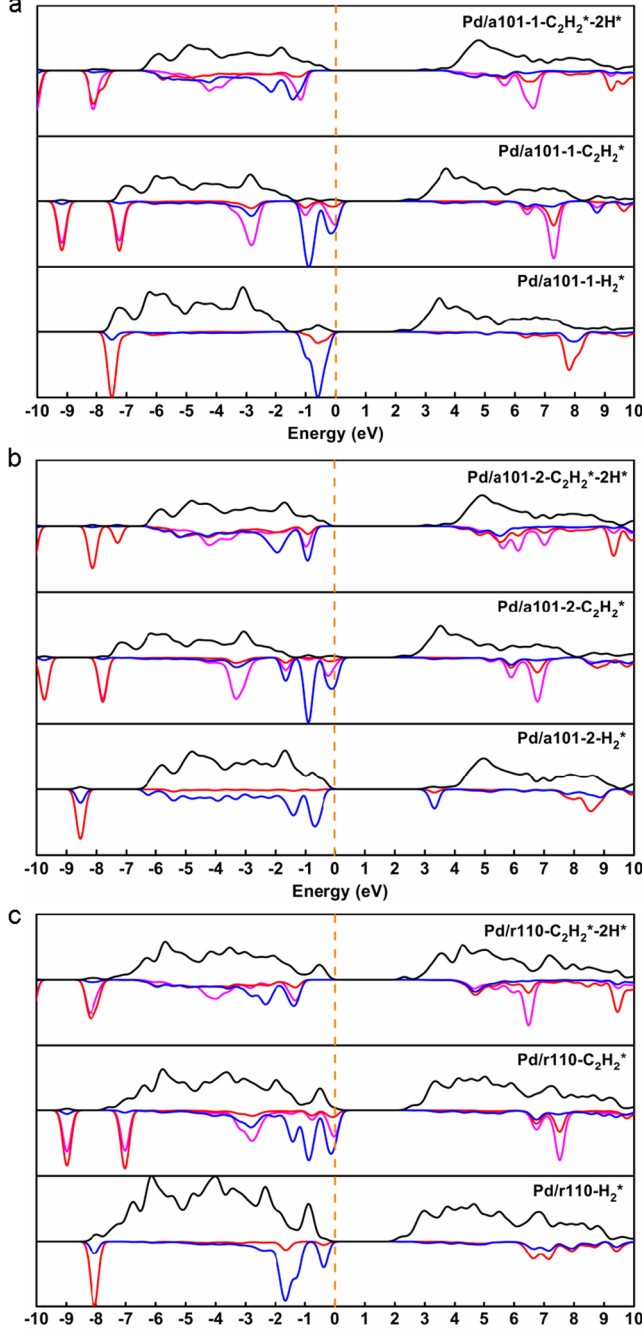

**Fig. 5 Projected density of states (PDOS) of ground states of nine adsorbed structures on catalysts. a** Pd/TiO₂-a101-1. **b** Pd/TiO₂-a101-2. **c** Pd/TiO₂-r110. Density of states that projected onto different kinds of atoms are drawn with different color: C: pink, H: red, Pd: blue, Total DOS: black. Fermi level has been shifted to 0 eV and marked by orange dashed line.

photo-thermo catalysis is lower than that in thermal catalysis (22.9 vs. 33.3 kJ mol⁻¹), Fig. 4e, implying that a different reaction pathway under light irradiation compared with the traditional thermal catalysis. Significantly, comparing with the previously reported catalysts, the Pd/TiO₂-600H is among the best and outperforms most Pd-based catalysts for semi-hydrogenation of acetylene including both NPs and SACs, Supplementary Table 1.

Further study was performed to reveal the underlying reason of the improved performance by photo-thermo catalysis process. In general, metals supported on semiconductor can serve to suppress electron–hole recombination and accept the photo-generated electrons via Schottky-junction. XPS measurement reveals clearly that after light irradiation the binding energy (B.E.) of Pd $3d_{5/2}$ shifted from 335.1 eV of Pd⁰ (metallic state) to 334.7 eV, Fig. 4f, indicating that photogenerated electrons in the conduction band of TiO₂ transferred to the adjacent Pd species to increase their electron density. It was proposed that electron-rich Pd is more effective for unsaturated hydrocarbons activation[75,76] and/or hydrogen activation[29]. To verify the possible facilitated hydrogen activation process, H₂-D₂ isotope exchange reaction was firstly performed on Pd/TiO₂-600H to evaluate H₂ dissociation ability upon light irradiation or not, Supplementary Fig. 14. The formation of HD could be observed immediately after D₂ pulse and its formation rate was almost the same under both cases. This excludes the improvement of H₂ dissociation by photo-thermo catalysis, implying that the main reason should be the improvement of acetylene activation.

**Simulation and calculation**. To verify our hypothesis that photogenerated carriers generated via excitation light facilitate the activation of acetylene, first-principle simulations were performed on a Pd₁/TiO₂ model catalyst. Note that in real catalyst, structures of reactive sites are not uniformed, and in our case, the support is a mixture of anatase and rutile. Detailed comparison between structures obtained from simulation and experiments are shown in the following Supplementary Table 2. Therefore, three kinds of widely reported structures were used to simulate the model catalyst (two for anatase, one for rutile, denoted as Pd/TiO₂-a101-1, Pd/TiO₂-a101-2 and Pd/TiO₂-r110, respectively, Supplementary Fig. 15)[77–79]. For each model, three kinds of adsorbed structures were considered, i.e., adsorbed H₂ (H₂*), adsorbed C₂H₂ (C₂H₂*), and C₂H₂*-dissociated hydrogen (C₂H₂*-2H*), based on the classical addition-elimination mechanism[44], Supplementary Fig. 16. Projected density of states (PDOS) analysis was performed on wavefunctions of adsorbed structure as shown in Fig. 5. Compared with the other two kinds of structures, in H₂-adsorbed structures, unoccupied states projected onto H element distribute at higher energy than those projected onto C element. This may imply it is harder to activate H–H bond than C–C bond, which is in agreement with the experiment results. According to PDOS, it is also shown that there are half-occupied states belong to C₂H₂ molecule in all C₂H₂-adsorbed structures but not in C₂H₂-2H-adsorbed structures. The half-occupied states mainly localized on C₂H₂ imply strong electron-donation effect of Pd single atom. In these cases, C–C bond is spontaneously activated. More DOS of adsorbed structures, especially about ethylene are provided in Supplementary Figs. 17–26.

To further study the effect of photo-induced activation of chemical bonds, in principle, classical excitation methods such as Time Dependent Density Field Perturbation Theory (TDDFPT), Complete Active Space Self Consistent Field (CASSCF) or multiconfigurational method should be used. However, for periodic system which contains hundreds of atoms (in our cases more than 150 atoms at most), it is less possible to calculate excitation states directly in a wide or even normal excitation energy range like isolated molecular system because numbers of states need to be calculated which increase exponentially respect to upper boundary of excitation energy chosen to consider, Supplementary Fig. 27. Fortunately, in our case, it is demonstrated that there must be electrons excited in charge-transfer type ways from orbitals localized to support or its relative part, to orbitals localized to absorbate, H₂ and/or C₂H₂. To simulate charge-transfer type excitation and obtain approximated wavefunctions of excited states, calculations based on Constraint Density Functional Theory (CDFT) were performed.

**Table 1 Bond orders of states and energy differences between charge-transfer states and their corresponding ground states in wavelength units.**

| Structure | Method | B.O. | λ/nm | Activation or not |
|---|---|---|---|---|
| $Pd/TiO_2$-a101-1-$C_2H_2$* | DFT (ground state) | 2.24 | – | – |
|  | CDFT Method 1 | 2.18[a] | 348.61[a] | Yes[a] |
|  | CDFT Method 2 | 2.25 | 487.22 |  |
| $Pd/TiO_2$-a101-1-$C_2H_2$*-2H* | DFT (ground state) | 2.22 | – | – |
|  | CDFT Method 1 | 2.25 | 884.29 |  |
|  | CDFT Method 2 | 2.16[a] | 500.58[a] | Yes[a] |
| $Pd/TiO_2$-a101-1-$H_2$* | DFT (ground state) | 0.65 | – | – |
|  | CDFT Method 1 | 0.63[c] | 321.88[c] | Negligible[c] |
|  | CDFT Method 2 | 0.70/0.74[b] | 886.71/286.78[b] | No[b] |
| $Pd/TiO_2$-a101-2-$C_2H_2$* | DFT (ground state) | 2.22 | – | – |
|  | CDFT Method 1 | 2.24 | 414.84 |  |
|  | CDFT Method 2 | 2.25 | 546.59 |  |
| $Pd/TiO_2$-a101-2-$C_2H_2$*-2H* | DFT (ground state) | 2.29 | – | – |
|  | CDFT Method 1 | 2.31 | 1420.39 |  |
|  | CDFT Method 2 | 2.24[a] | 541.86[a] | Yes[a] |
| $Pd/TiO_2$-a101-2-$H_2$* | DFT (ground state) | 0.68 | – | – |
|  | CDFT Method 1 | 0.67[c] | 315.75[c] | Negligible[c] |
|  | CDFT Method 2 | 0.74/0.76[b] | 594.23/285.80[b] | No[b] |
| $Pd/TiO_2$-r110-$C_2H_2$* | DFT (ground state) | 2.22 | – | – |
|  | CDFT Method 1 | 2.21[a] | 479.17[a] | Yes[a] |
|  | CDFT Method 2 | 2.26 | 563.57 |  |
| $Pd/TiO_2$-r110-$C_2H_2$*-2H* | DFT (ground state) | 2.29 | – | – |
|  | CDFT Method 1 | 2.32/2.25[a] | 3272.00/661.10[a] | Yes[a] |
|  | CDFT Method 2 | 2.35/2.15[a] | 863.89/337.65[a] | Yes[a] |
| $Pd/TiO_2$-r110-$H_2$* | DFT (ground state) | 0.60 | – | – |
|  | CDFT Method 1 | 0.61 | 381.44 |  |
|  | CDFT Method 2 | 0.66/0.69[b] | 699.54/241.22[b] | No[b] |

Excitation wavelengths of 18 possible modes on 9 structures are obtained by performing further CDFT by increasing number of electrons transferred, for more details, see the Supplementary Methods, Supplementary Fig. 29 and Supplementary Figs. 32–34. $TiO_2$-a101: anatase (101), $TiO_2$-r110: rutile (110). For more detailed structural information, see the Supplementary Figs. 15, 16. Although there are modes showing activation of unsaturated C–C chemical bond with wavelength larger than 480 nm, which may have negligible contribution to the activity improvement upon irradiation light in our case.
[a]Modes showing decrease in B.O. and their wavelengths are larger than 320 nm are marked as "yes".
[b]Modes showing increase in B.O. and wavelengths shorter than 320 nm are marked as "no".
[c]Modes showing small decrease in B.O. but wavelengths close to 320 nm are marked as "negligible".

CDFT is the method that extends energy functional by adding more variational terms (constraints), then optimizes extended energy functional as an outer loop of regular self-consistent-field calculation. Works on excitation properties with help of CDFT have been widely reported and discussed in depth[80,81]. Those constraints are always constructed manually with reasonable consideration. In our case, the number of electrons transferred was considered and tuned in two ways (from $TiO_2$ to Pd-adsorbate composite (Method 1) or from catalyst (Pd/$TiO_2$) to adsorbate (Method 2), see Computational details). The more electrons transfer to anti-bonding orbitals of adsorbate to activate chemical bonds (H–H or C–C), the higher excitation energy (the shorter wavelength of light used to excite electron) will be needed, Supplementary Fig. 28. Therefore, possibility of charge-transfer type excitation caused activation of chemical bonds can be estimated by judging if corresponding wavelength can locate in the catalyst absorption wavelength range. Wavefunctions of charge-transferred states were optimized with PBE0-TC-LRC hybrid functional, based on that of ground state also with ground state optimized geometry, then energy differences between those of two kinds of states were calculated and treated as approximations of vertical excitation energies. Mayer bond orders (B.O.) were calculated as scalar descriptors to estimate impacts of charge transfer on chemical bonds activation listed in Table 1[82].

As shown in Supplementary Fig. 28, when increasing number of electrons transferred, at first, electrons will fill orbitals with lower energy which contain components of bonding orbitals of adsorbate (significant in PDOS of $C_2H_2$-adsorbed structures shown in Fig. 5), so there will be small increase in bond order.

Then electrons start to fill orbitals with higher energy which contains anti-bonding orbitals of adsorbate, so bond order shows rapid decrease. However, in real excitation process, there is no need or it is not compulsory to fill energy-lower orbitals first, so energy required for excitations which have activation effect on chemical bonds will only be smaller but not larger, i.e., wavelength will only be longer but not shorter. In our CDFT simulated system, it turned out that C–C activation indeed exists in Pd/$TiO_2$-a101-1-$C_2H_2$*-2H*, Pd/$TiO_2$-a101-2-$C_2H_2$*-2H*, Pd/$TiO_2$-r110-$C_2H_2$* and Pd/$TiO_2$-r110-$C_2H_2$*-2H* models at least, H–H activation does not exist in all our models which shows consistency with ground state PDOS, and in good agreement with our experiments again. There are some excitation modes left uncertain in Table 1, but it is already enough to summarize that light irradiation can improve the reactivity of acetylene semi-hydrogenation by facilitating acetylene activation. A possible mechanism as well as the reaction process is proposed in Supplementary Fig. 30, described as following: (1) Irradiation leads to band-gap excitation of $TiO_2$ support, generating electron–hole pairs, (2) $H_2$ dissociation goes to a heterolytic pathway on isolated Pd sites and $TiO_2$ (at the Pd–O interface), (3) $C_2H_2$ molecules adsorb on isolated Pd sites and then photo-generated electrons transfer to isolated Pd sites, facilitating the activation of adsorbed $C_2H_2$, and (4) H species react with the activated $C_2H_2$ on Pd single atoms to form Pd-$C_2H_4$ species, followed by $C_2H_4$ desorption. According to this mechanism, there should not be net charge accumulation. However, this seems in contrast to the XPS characterization where net charge accumulation occurs upon irradiation. It is conjectured that without the

presence of reactants, a certain degree of net charge accumulation occurs because Pd atoms can accept some electrons from $TiO_2$, resulting in a difficulty in electron–hole recombination. However, under reaction condition electrons can be consumed by reactants, inhibiting the continuous net charge accumulation. This can be verified by the stability of a 0.1 wt% $Pd/TiO_2$-NP-200H catalyst (to avoid any plausible effect of light on the SMSI state). It can be seen that after an evaluation in the initial few hours in dark condition, the catalytic performance is quite stable upon light irradiation, excluding the continuous net charge accumulation (which may arouse activity change), Supplementary Fig. 31.

## Discussion

In summary, we employed a simple yet general method to dramatically improve the selectivity of $Pd/TiO_2$ catalysts in semi-hydrogenation of acetylene via selectively encapsulating the co-existed small amount of Pd clusters/NPs to construct $Pd_1/TiO_2$ SACs according to their different SMSI occurrence conditions. Moreover, the photo-thermo catalysis was applied to this process and a much-improved catalytic activity was obtained: Upon photo-thermo catalysis condition the semi-hydrogenation of acetylene can be realized at temperature as low as 70 °C. Detailed studies combined with DFT calculation revealed that photo-induced electrons transferred from $TiO_2$ to the adjacent Pd species facilitate the activation of acetylene thus benefiting the reaction process. This work offers a promising strategy to manipulate the catalytic performance of acetylene semi-hydrogenation, and may also open up a window for photo-thermo catalytic selective hydrogenation reaction.

## Methods

**Materials and chemicals**. Palladium phthalocyanine (PdPc, 98%) was purchased from 3A Materials, Titanyl phthalocyanine (TiOPc, purified by sublimation) was purchased from Tokyo Chemical Industry (TCI), and ethanol (99.7%) was purchased from Sinopharm Chemical Reagent Co. Ltd. Ammonia (25 wt %) and Polyvinyl alcohol (PVA, Mw = 10,000, 80% hydrolyzed) were purchased from Aladdin. Titanium (IV) oxide (mixture phases (99.7%) and rutile (AR)) and sodium borohydride (NaBH₄, 98%) were purchased from Alfa Aesar. Tetra-amminepalladium (II) nitrate solution ($Pd(NO_3)_2 \cdot 4NH_3$, 10 wt% in $H_2O$) and Sodium tetrachloropalladate (II) ($Na_2PdCl_4$, Pd 30%) were purchased from Sigma-Aldrich. All reagents were used without further purification. Deionized water was obtained from a Millipore Autopure system.

## Catalyst preparation

*Ball milling.* All in situ milling experiments were performed in Planetary Ball Mill of RETSCH PM100 with a $ZrO_2$ capsule (V = 50 ml) and $ZrO_2$ milling balls (15 × 5 mm and 15 × 3 mm), and the milling speed was 400 rpm during the whole process. For the preparation of support $TiO_2$, TiOPc (4 g) was filled in the chamber firstly by a brief ball-milling process for 20 min, then 10 mL ethanol was added for another 12 h grinding. The mixture was subsequently filtered and dried at 60 °C overnight, and the obtained material was calcined at 600 °C for 2 h in air atmosphere.

For the preparation of $Pd/TiO_2$, phthalocyanine (Pc) precursors containing metal centers for Pd and Ti, i.e., PdPc (5 mg) and TiOPc (4 g) were mixed with a low weight ratio of 1: 800 (i.e., "precursor-dilution" strategy) firstly by a brief ball-milling process for 20 min, then 10 mL ethanol was added to the chamber for another 12 h grinding. The mixture was filtered and dried at 60 °C overnight. After this process, PdPc was dispersed in the bulk TiOPc. After calcination at 600 °C for 2 h in air, the two precursor salts were converted to $PdO_x$ and $TiO_2$, respectively, and Pd species was anchored on the support $TiO_2$ in the form of single atoms together with small amount of Pd NPs. The synthesized catalyst was denoted as $Pd/TiO_2$. It was followed by reducing under 10 vol% $H_2/He$ gas at 200 °C, 600 °C for 0.5 h, denoted as $Pd/TiO_2$-200H and $Pd/TiO_2$-600H, respectively. The $Pd/TiO_2$-600H catalyst was re-oxidized under 10 vol% $O_2/He$ at 300 °C for 0.5 h, denoted as $Pd/TiO_2$-600H-O300.

*Incipient wet impregnation.* 0.15 wt% $Pd/TiO_2$ catalyst was synthesized by incipient wet impregnation of 1 g $TiO_2$ (mixture phases or rutile) with 1.5 mL 1 mg mL⁻¹ solution of sodium tetrachloropalladate (II) and 3.5 mL $H_2O$. Prior to impregnation, the support was pre-reduced at 250 °C for 1 h (20 vol% $H_2/Ar$, 50 mL min⁻¹). After impregnation, the sample was dried at 60 °C and then calcined in a muffle furnace at 450 °C for 4 h (denoted as $Pd/TiO_2$-IWI). Pd loading of $Pd/TiO_2$-IWI determined by ICP was 0.146 wt%. It was followed by reducing under 10 vol% $H_2/$

He gas at 200 °C and 650 °C for 0.5 h, denoted as $Pd/TiO_2$-IWI-200H and $Pd/TiO_2$-IWI-650H, respectively.

*Strong electrostatic adsorption (SEA).* As previously described[70], 25 mL of deionized water was mixed with 75 mL of $NH_4OH$ to dissolve 1 g $TiO_2$ support. Separately, 40 μL of $Pd(NO_3)_2 \cdot 4NH_3$ was added to 25 mL of $NH_4OH$ to stir and disperse. The 25 mL $Pd(NO_3)_2 \cdot 4NH_3$ solution was injected dropwise into the support solution over 2 h by a constant flow pump stirring constantly. After precursor addition, the stirring solution was heated to 70 °C until completely dried. The dried catalyst was calcined in a muffle furnace at 450 °C in air for 4 h (denoted as $Pd/TiO_2$-SEA). Pd loading of $Pd/TiO_2$-SEA determined was 0.149 wt%. It was followed by reducing under 10 vol% $H_2/He$ gas at 200 °C and 650 °C for 0.5 h, denoted as $Pd/TiO_2$-SEA-200H and $Pd/TiO_2$-SEA-650H, respectively.

*Sol–gel method.* 0.1 wt% $Pd/TiO_2$ NP catalyst was synthesized by sol–gel method. Typically, 0.835 mL of PVA solution (2 mg mL⁻¹) and 1 mL of $Na_2PdCl_4$ solution (2 mg mL⁻¹) were added to 10 mL aqueous solution at room temperature under vigorous stirring. After 30 min, 1.4 mL of 2 mg mL⁻¹ NaBH₄ solution was rapidly injected into the solution which turn to dark immediately, indicating the formation of Pd colloid. 2 g of support $TiO_2$ dispersed in 50 mL aqueous solution was then added and the mixture was continuously stirred for 12 h. The solid was collected by filtration, washed with deionized water, dried at 60 °C overnight, and then calcined at 400 °C for 5 h under air. The obtained sample was then reduced in $H_2/He$ (10 vol% $H_2/He$, 30 ml · min⁻¹) at 200 °C for 0.5 h prior to the hydrogenation reaction, denoted as $Pd/TiO_2$ NP-200H.

## Characterizations

*BET.* Brunauer–Emmett–Teller (BET) surface area of $TiO_2$ as well as the supported catalysts was measured by $N_2$ adsorption at 77 K using a Micromeritics ASAP 2010 apparatus. The samples were degassed at 100 °C for 1 h and 300 °C for 4 h before nitrogen adsorption.

*UV–Vis spectra.* UV–Vis spectra were collected by a UV/Vis/NIR spectrophotometer (Perkin-Elmer, Lambda 950) with Peltier-controlled PbS detector. The data interval is 0.2 nm and UV/Vis bandwidth is 2 nm.

*ICP-AES/OES.* The actual Pd loadings of all catalysts were determined by inductively coupled plasma atomic emission spectroscopy (ICP-AES) on an IRIS Intrepid II XSP instrument (Thermo Electron Corporation) or inductively coupled plasma Optical Emission Spectrometer (ICP-OES) on ICPS-8100 instrument (Shimadzu Co., Ltd.).

*XRD.* Powder XRD was performed at a PANalytical X'Pert PRO X-ray diffractometer using Cu-Kα radiation ($\lambda$ = 0.15432 nm), operating at 40 kV and 40 mA.

*HAADF-STEM, AC-HAADF-STEM, and EELS.* The high-angle annular dark-field scanning transmission electron microscopy (HAADF-STEM) images were obtained on JEOL JEM-2100F operated at 200 kV. The aberration-corrected high-angle annular dark-field scanning transmission electron microscopy (AC-HAADF-STEM) images and the electron energy loss spectroscopy (EELS) were obtained on JEOL JEM-ARM200F operated at 200 kV equipped with a Gatan Quantum 965 image filter system. The chemical compositions of the covering layer of Pd NPs were characterized by directly putting electron beam at the Pd NPs in a STEM mode. Before measurements, the samples were ultrasonically dispersed in ethanol, and then a drop of the solution was put onto the carbon film supported by a copper grid.

*DRIFTS.* In situ diffuse reflectance infrared Fourier transform spectra (DRIFTS) were acquired with a VERTEX 70 V infrared spectrometer equipped with a mercury cadmium telluride (MCT) detector and operated at a resolution of 4 cm⁻¹ using 32 scans. Before CO adsorption, all catalysts except $Pd/TiO_2$-fresh were reduced at corresponding temperatures with 10 vol% $H_2/He$ and purged with pure He for 30 min at the same temperature before being cooled. After cooled to room temperature, the background spectrum was recorded and then 2 vol% CO/He was introduced into the reaction cell, and the spectra were collected until the state steady. Subsequently, pure He was introduced again to desorb the gas phase CO, and the spectra were also recorded.

*XANES.* The X-ray absorption spectroscopy study was performed at the BL08B2* of SPring-8 (8 GeV, 100 mA), Japan, in which the X-ray beam was monochromatized with water-cooled Si (111) double-crystal monochromator and focused with two Rh coated focusing mirrors with the beam size of 2.0 mm in the horizontal direction and 0.5 mm in the vertical direction around sample position, to obtain X-ray adsorption fine structure (XAFS) spectra both in near and extended edge.1 Pd foil and PdO samples were used as references. The catalysts and standard samples were measured with fluorescence and transmission mode, respectively. The spectra were analyzed and fitted using an analysis program Demeter.2 Before measurement, the $Pd/TiO_2$ sample was reduced under 10 vol% $H_2/He$ at 200 and

600 °C for 0.5 h, followed by purging with pure He until cooled to room temperature.

*XPS.* X-ray photoelectron spectroscopy characterization was conducted on an X-ray photoelectron spectrometer (USA, ThermoFischer, ESCALAB 250Xi) equipped with Al Kα excitation source (1486.8 eV) with C as the internal standard (C 1s = 284.8 eV). Before measurement, the $Pd/TiO_2$ sample was reduced under 10 vol% $H_2$/He at 200 °C for 0.5 h, followed by purging with pure He until cooled to room temperature.

*MS.* The $H_2$-$D_2$ isotope exchange experiments were performed in the reactor combined with an online mass spectrometer (OmniStarTM, Pfeiffer Vacuum). 5 mg catalyst of $Pd/TiO_2$ diluted with 400 mg quartz sand was placed in the quartz reaction chamber, reduced at 600 °C under 10 vol% $H_2$ flow for 0.5 h, and purged with He to room temperature. Then $H_2$-$D_2$ exchange experiments were conducted with light off at 65 °C and light on at 65 °C (including light induced heating effect) by Xe lamp, respectively. Typically, the $H_2$/He mixture (1: 1) was flowed until stable and then $D_2$ pulse was sent into the reactor ($H_2$: He: $D_2$ = 1: 1: 0.5), and repeated this intake process 4 times.

**Catalytic performance and stability evaluation.** Acetylene semi-hydrogenation in excess ethylene over different $Pd/TiO_2$ catalysts was evaluated in a quartz fixed-bed flow reactor (d = 10 mm) with 15 mg catalysts diluted by 200 mg quartz sand. The as-prepared catalysts were reduced in $H_2$ (10 vol% $H_2$/He, 30 ml min$^{-1}$) at 200 °C and 600 °C for 0.5 h prior to the hydrogenation reaction. After being cooled to room temperature, feed gas of 1 vol% $C_2H_2$, 10 vol% $H_2$, 20 vol% $C_2H_4$, and balance He (30 ml min$^{-1}$) was introduced to the fixed-bed reactor followed by temperature programmed testing. The reaction temperature was held constant for 20 min before ramping to the next temperature point. Gas composition at the inlet and outlet were analyzed by online GC (A91) equipped with a flame ionization detector (FID) and a PORAPAK-N column with helium as the carrier gas. From our results, $C_2H_4$ and $C_2H_6$ were the only $C_2$ products while the formation of oligomers was negligible.

The photo-thermo catalysis of acetylene selective hydrogenation in excess ethylene was conducted in a quartz fixed-bed flow reaction chamber equipped with a quartz window (d = 35 mm) on the top side for light introduction. Xe lamp with adjustable power (Beijing PerfectLight, PLS-SXE300) was introduced to radiate light. Light power density on the catalyst thin layer was tested by an optical power meter (THOR LABS PM400 Optical Power Meter). The potential heat effect was minimized by using a water bath equipped with circulating water. After various measurements in control experiment, our oft-repeated experiments suggested that the light can only increase the reactor temperature by 10 °C at reaction temperature below 100 °C. As a result, a 10 °C-correction was used when performing photo-thermo catalysis. A certain amount of $Pd/TiO_2$ catalyst (or 0.036 wt% $Pd/Al_2O_3$ catalyst sample) diluted by 400 mg quartz sand was uniformly dispersed as a thin layer. The thin layer catalyst was in situ reduced in $H_2$/He (10 vol% $H_2$/He, 30 ml min$^{-1}$) at 600 °C for 0.5 h prior to the hydrogenation reaction. After being cooled to room temperature under purging with He, 30 ml min$^{-1}$ feed gas of 1 vol% $C_2H_2$, 10 vol% $H_2$, 20 vol% $C_2H_4$ and He was introduced to the fixed-bed reactor. $C_2H_4$ and $C_2H_6$ were the only $C_2$ products while oligomers formed could be ignored.

The conversion of acetylene and selectivity to ethylene were calculated as Eq. 1 and Eq. 2:

$$\text{Conversion} = \frac{\text{C2H2(feed)} - \text{C2H2}}{\text{C2H2(feed)}} \times 100\% \tag{1}$$

$$\text{Selectivity} = \left(1 - \frac{\text{C2H6} - \text{C2H6(feed)}}{\text{C2H2(feed)} - \text{C2H2}}\right) \times 100\% \tag{2}$$

## Data availability
The data that support the findings of this study are available within the paper and its Supplementary information, and all data are available from the authors on reasonable request.

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

## Acknowledgements

The authors thank Prof. Can Li at Dalian Institute of Chemical Physics for his valuable discussions and suggestions. This work was financially supported by National Key Research and Development Program of China (2021YFA1500503), National Natural Science Foundation of China (21972135, 21961142006, 22090033 and 51701201), and CAS Project for Young Scientists in Basic Research (YSBR-022). The synchrotron radiation experiment for XAFS was performed at the BL08B2 of SPring-8 with the approval of Japan Synchrotron Radiation Research Institute (Proposal Nos. 2019B3415, 2020A3415).

## Author contributions

B.Q. and T.Z. conceived and supervised the project. Y.G. carried out the catalyst synthesis, catalytic performance test, stability evaluation, and conducted some

characterizations. Y.H. conducted the simulation and calculation. B.H., M.A., Q.L., and Li.Li. helped to analyze the data. Y.S. and Q.J. carried out the STEM and EELS characterizations. Y.T.C. and Le.Li. carried out the XAS measurements and analysis. B.Z., M.S., and Y.Z. performed some photocatalytic experiments and helped on the discussion of mechanism. R.L. guided and supervised the experiments of photo-thermo catalysis and wrote the corresponding section. Y.G. and B.Q. wrote the paper. All authors contributed to project discussions and modified the paper.

## Competing interests

The authors declare no competing interests.
