## [Peer Review File · Nature Communications]

Title: Photo-thermo semi-hydrogenation of acetylene on Pd1/TiO2 single-atom catalystREVIEWER COMMENTS

Reviewer #1 (Remarks to the Author):

In the article entitled "Photo-thermo semi-hydrogenation of acetylene on Pd1/TiO2 single-atom catalyst", the authors present convincing experimental and theoretical evidences of semi-hydrogenation of acetylene improved by the combined effect of the presence of Pd Single Atom Catalysts and formation of electron-hole pairs in the oxide support. Clearly it is a significant work, with a methodology that is well founded. It is not entirely original, in the sense that photo-thermal catalysts have been already used for others reactions such as CO2 reduction.

The concept of Strong Metal-Support Interaction is thus evoked as the main mechanism to selectively control the amount small NP/large NP of Pd grown on the support. By the way, historical references on this point should be added in the introduction to my opinion.

The description of the synthesis procedure as well as the computational details allow for work to be reproduced. I think the present paper could be published after considering the following points:

-- what is the mechanism of formation of the SAC. It is not clear to me which step of the synthesis is responsible for their presence, and which experimental parameter(s) would be able to control the NP/SAC ratio ?

--To be entirely convincing on the nature of the Pd-1 catalytic site could the authors provide information such as XPS data that usually give hints about the charge character on the metal center ?

-- In the same spirit, if the authors could provide Density of States (DoS) pictures out of DFT calculations it would certainly help the readers to better understand the electronic structure of those SACs.

-- A more technical details on the DFT models. Do the authors have to apply for dipolar correction on the support cell to estimate properly the energy ? It is not obvious that polarization effect upon Pd adsorption can be neglected.

-- There are several reports in the literature now, proving experimentally and theoretically that H2 can not dissociate easily on SAC, but rather on metal NPs. I think that fact deserves an additional comment.

--Thermodynamics aspects of the C2H2 and H2 adsorption on Pd1 site are missing in the discussion. Could the authors provide such information ?

-- a minor comment on the SI:

Table X1 would be more valuable if errors were given instead.

Sincerely yours,

Reviewer #2 (Remarks to the Author):

In this article, Li, Qiao et al report important results on the selective hydrogenation of acetylene, an important reaction in industry. The authors are based on convincing experimental results, supported by theoretical calculations to demonstrate a photo-thermal effect operating on ultradisperse Pd/TiO₂ catalysts. The significance of these results is in line with a publication in Nat. Commun. once the following changes have been introduced:

1) The proposed green chemistry approach should be discussed in the perspective of the following article: Room-temperature electrochemical acetylene reduction to ethylene with high conversion and selectivity. Nat Catal 4, 565–574 (2021). <https://doi-org.inc.bib.cnrs.fr/10.1038/s41929-021-00640-y>
2) the authors use as support a rutile/anatase mixture, which is certainly not innocent (cf. J. Phys. Chem. B 2006, 110, 15, 8019–8024; J. Chem. Phys. 136, 104107 (2012)). It is not clear in the manuscript whether the PdSAs are localized on the rutile and/or anatase phase. Please use the EELS analyses to try to see more clearly (cf 10.1557/proc-139-327). Please also comment on the binding energies of the PdSAs on these two phases. In order to be able to conclude on the nature of the active phase PdSA/TiO₂-a or PdSA/TiO₂-r.

Reviewer #3 (Remarks to the Author):

In this work, Guo et al. demonstrated semi-hydrogenation of acetylene in excess ethylene on atomically dispersed Pd species on TiO₂ supports prepared by two different methods. The authors exploited the strong metal-support interaction (SMSI) to selectively encapsulate Pd nanoparticles and clusters by the SMSI overlayer via high-temperature reduction of the catalysts in H₂, leaving single-atom Pd species exposed for catalysis. This treatment resulted in substantial improvement in C₂H₄ product selectivity in thermocatalysis. The selectivity gradually dropped from 60% to about 40% over 40 hours of reactivity test with stable %100 C₂H₂ conversion. Next, the authors demonstrated that high energy near UV photons could be utilized to drive the same reaction at milder operating conditions. Supported by their computational and experimental studies, the authors suggest that the origin of the reactivity enhancement in photocatalysis is the photoexcitation of TiO₂ and the transfer of charge carriers to Pd sites for chemical bond activation (primary to activate C-C bonds). The combination of the SMSI and light to control catalysis is interesting! However, there are significant concerns on the photocatalysis side and the associated discussion. In particular, an important aspect of the photocatalysis results that the authors seemingly refused to acknowledge is regarding the selectivity of the process. Light can promote the reactivity but also lead to a substantial drop in desired product selectivity in this case (see below). This poses a major concern for using the current photocatalysis approach for driving the semi-hydrogenation of acetylene on the proposed catalyst system.

- Comparison of the results in Figure 4 and S8 (e.g., focusing on their panel (a)) reveals that light-promoted reactivity comes at the expense of a substantial drop in the C₂H₄ product selectivity. Looking

at the entire 35 min of the experiment, the selectivity decreases from 80% to near 40%. Even with the assumption that some drop in selectivity during the initial light-on cycle is due to measurement under non-steady-state condition (that has its own issue), it is clear that there is an additional 10% drop in the selectivity each time that light is turned on with a total of near 20% selectivity drop just in two illumination cycles. Regrettably, the authors consider this as a “slight” decrease in selectivity or even “stable” selectivity (during the second cycle)!

- The drop in selectivity also implies that additional light-on cycles (if performed) could potentially lead to a further decrease in selectivity.
- Furthermore, the light-induced drop in the C₂H₄ product selectivity is an important result that is deserved to be highlighted in the main text along with the reactivity results in photocatalysis.
- Notably, the authors repeatedly state that there is only one report on heterogeneous photocatalytic semi-hydrogenation of acetylene (ref. 26, Zhou et al., Adv Mater 2019) while oversight the previous original work in this area (Swearer et al., PNAS 2016) using visible light and a plasmonic photocatalyst for semi-hydrogenation of acetylene. Interestingly, unlike the current results, the work by Swearer et al. demonstrated that light could promote both the reactivity and the selectivity for this reaction.
- The authors should also discuss how the use of light in this system leads to worsened C₂H₄ selectivity. Reportedly, the results of DFT calculations and H₂-D₂ exchange experiment suggest the primary role of Pd/TiO₂ photoexcitation being the C-C bond activation acetylene activation, but the discussion of product selectivity is missing.
- Regarding the authors' discussion on the role of light and the underlying mechanism: It is stated that the reactivity enhancement is due to a charge-driven mechanism from photoexcitation of TiO₂ and as supported by the authors XPS and DFT results. This statement is broadly true. However, there are important aspects of light-induced catalyst heating that should be considered when discussing the role of light in photocatalytic processes. For instance, despite the authors' statement, the use of a circulating water bath does not necessarily exclude the photothermal heating effect in catalysis. This is due to the potential presence of local nanoscale hot spots, especially with supported metal catalysts. Also, the magnitude of the light-induced heating within the catalyst bed strongly depends on multiple factors, including the optical properties (ensemble photon absorption) and thermal conductivity of the catalyst, incident photon flux, illumination wavelength, etc. Currently, the information regarding the power density of the incident photons in Figure 4 is missing. Also, the use of Al₂O₃ support in the control experiment and the lack of photocatalytic reactivity does not necessarily exclude the thermal effect on TiO₂ since this is a comparison of two different systems with a very different absorption properties and thermal conductivities.
- Furthermore, the authors state that “it is suggested that the light can still increase the reactor temperature by 10 oC below 100 oC. Therefore, a 10 oC-correction was used when performing photo-thermo catalysis.” Suggested by who and based on what evidence? Again, the magnitude of light-induced heating depends on several factors, as discussed above. Also, did the authors apply this “undocumented” correction to the Arrhenius plot of photocatalysis in Figure 4e?
- The authors suggested a charge carrier-driven mechanism for bond activation under illumination with charge transfer from TiO₂ to Pd sites. They should also provide a scheme to discuss the proposed mechanism further, especially to examine whether there is a net charge transfer process to adsorbate(s) and that the catalyst remains in a neutral state or not. If the process is associated with net charge

transfer, it results in a charge build-up and eventually a decrease in photocatalytic reactivity. A long-term stability test in photocatalysis should therefore be provided.

- Spectrum of the light source used for illumination is missing.
- Intensity-dependent photocatalytic studies should provide more insights regarding light-matter interactions for photocatalysis in this system?
- Improper citations! The past decade has witnessed huge progress in light-driven heterogeneous catalysis, especially with supported metal catalysts, using a combination of light and external heating or only light to drive the chemistry (the latter is more ideal). The authors must more appropriately acknowledge the previous efforts in this area rather than citing a few very recent examples as the pioneering work in the field, which is indeed an incorrect remark!!!
- The authors consider photogenerated carriers in TiO₂ as “hot carriers”. This should not be confused from well-known hot carrier generation in plasmonic metal nanostructure.
- Words such as “obvious”, “clearly”, etc. are overused throughout the manuscript. Their use should be minimized as they do not necessarily always reinforce the best scientific scenario.

Respond to Reviewers

We appreciate all the reviewers for their encouraging comments, valuable questions and constructive suggestions which are helpful to improve our manuscript. Specific response/answers to each question/suggestion are listed below.

Response to Reviewer 1

In the article entitled "Photo-thermo semi-hydrogenation of acetylene on Pd₁/TiO₂ single-atom catalyst", the authors present convincing experimental and theoretical evidences of semi-hydrogenation of acetylene improved by the combined effect of the presence of Pd Single Atom Catalysts and formation of electron-hole pairs in the oxide support. Clearly it is a significant work, with a methodology that is well founded. It is not entirely original, in the sense that photo-thermal catalysts have been already used for others reactions such as CO₂ reduction.

The description of the synthesis procedure as well as the computational details allow for work to be reproduced. I think the present paper could be published after considering the following points:

1--The concept of Strong Metal-Support Interaction is thus evoked as the main mechanism to selectively control the amount small NP/large NP of Pd grown on the support. By the way, historical references on this point should be added in the introduction to my opinion.

Response: We truly thank the reviewer for this good suggestion. We have mentioned the concept of SMSI in introduction section in our revised manuscript and cited corresponding references: Strong metal-support interaction (SMSI), a topic being extensively studied for more than 40 years in heterogeneous catalysis area, [*Angew. Chem. Int. Ed.*, 2020, 59, 11824-11829; *Nat. Commun.*, 2020, 11, 5811; *Acta Phys.-Chim. Sin.*, 2022, 38, 2101039; *ACS Catal*, 2017, 7, 7461-7465; *Angew. Chem. Int. Ed.*, 2015, 127, 4627-4631] has sparked renewed interests due to their potential in modifying catalyst performance, and especially in stabilizing catalysts. [*Angew. Chem. Int. Ed.*, 2016, 55, 10606-10611; *J. Catal.*, 2015, 330: 19-27] Recently, we found that isolated Pt atoms supported on TiO₂ can manifest classical SMSI [*Angew. Chem. Int. Ed.*, 2020, 59, 11824-11829] but at a much higher reduction

temperature. The feature of this finding is that the co-existed nanoparticles (NPs) can be selectively encapsulated while single atoms keep exposed through reduction at suitable temperatures. This finding might be extended to TiO₂ supported other metal catalysts thus providing a new strategy to construct stable SACs.

2--What is the mechanism of formation of the SAC. It is not clear to me which step of the synthesis is responsible for their presence, and which experimental parameter(s) would be able to control the NP/SAC ratio?

Response: In this work, we have used three methods to prepare Pd/TiO₂ catalysts, i.e., ball-milling, strong electrostatic adsorption (SEA) and a common impregnation method. The first one is based on the “precursor-dilution” mechanism [*Nat. Commun.* 2019, 10, 3663] while the second mechanism is “*promoting attractive interaction between Pd ions and the support*” [*J Am Chem Soc.* 2017, 139, 14150-14165] for SAC preparation. Our original purpose was to prepare “absolute” Pd single-atom catalysts (SACs), i.e., the presence of only isolated single atoms without any clusters/NPs in our catalysts. However, the results showed that even using such effective methods the presence of Pd nanoparticles/clusters is still unavoidable, highlighting the difficulty of preparing absolute Pd single-atom catalysts [*Chem. Rev.* 2020, 120, 11986-12043]. This is the reason why we’d like to try to encapsulate the co-existed Pd NPs by utilizing the different strong metal-support interaction (SMSI) concurrence conditions of Pd single atoms and Pd NPs. We think this is one of the major values/significances of this work, that is, it provides a simple yet relatively universal strategy to “tune” common supported Pd catalysts into Pd SACs with good reversibility. The impregnation method was designedly used to strengthen the value of this strategy because impregnation method is simple and has been widely used in industrial process, which generally results in the co-existence of single atoms and NPs. The results turn out that Pd catalysts prepared by such a common and simple method can be “tuned” into Pd SACs, highlighting the universality and potential application of this strategy.

As to the ratio of Pd single atoms and NPs, we don’t think the ratio matters so we didn’t try to control; and it’s actually hard to control by the methods we adopted.

3--To be entirely convincing on the nature of the Pd-1 catalytic site could the authors provide information such as XPS data that usually give hints about the charge character on the metal center?

Response: Thank you for your good suggestion. In our original manuscript, the charge character of Pd had been characterized by XANES. It suggested that the Pd existed as positively charged state. Considering the fact that under SMSI state Pd NPs often existed as metallic or even negatively charged state [*Catal. Today.* 2016, 260, 21-31], the positive chemical state is attributed to Pd single atoms [*Chem. Rev.*, 2020, 120(21):11986-12043; *Nanoscale*, 2017, 9, 6643-6648; *J. Am. Chem. Soc.* 2018, 140, 954-962]. It is also suggested that the proportion of Pd NPs in the catalyst should be low as their metallic or negatively charged state can be compensated by the positively charged state of Pd single atoms. According to the review's suggestion, we further performed XPS measurement (Figure R1 below). The binding energies of Pd 3d can be fitted into two peaks of 335.1 eV and 336.5 eV even after reduction at 600 °C, which are assigned to Pd⁰ and Pd^{δ+}, respectively. This result is consistent with the XANES result and further verifying the high valence of Pd single atoms. We have added the result and discussion into the revised manuscript and supplementary file (Supplementary Fig. 5).

Figure R1. Pd 3d XPS of Pd/TiO₂-600H

On the other hand, we also calculated the Bader charges of Pd single atoms in different models in which Pd single atoms are all positively charged:

Table R1. Bader charges of Pd atoms in different models

Catalysts		Singly adsorbed structures		Co-adsorbed structures	
model	Q_{Pd}/e	model	Q_{Pd}/e	model	Q_{Pd}/e
Pd/a101-1	+0.20	Pd/a101-1-H ₂	+0.13	Pd/a101-1-2H-C ₂ H ₂	+0.49
Pd/a101-2	+0.16	Pd/a101-2-H ₂	+0.70	Pd/a101-2-2H-C ₂ H ₂	+0.48
Pd/r110	+0.30	Pd/r110-H ₂	+0.15	Pd/r110-2H-C ₂ H ₂	+0.42
		Pd/a101-1-C ₂ H ₂	+0.20	Pd/a101-1-2H-C ₂ H ₄	+0.48
		Pd/a101-2-C ₂ H ₂	+0.17	Pd/a101-2-2H-C ₂ H ₄	+0.47
		Pd/r110-C ₂ H ₂	+0.23	Pd/r110-2H-C ₂ H ₄	+0.40
		Pd/a101-1-C ₂ H ₄	+0.17		
		Pd/a101-2-C ₂ H ₄	+0.18		
		Pd/r110-C ₂ H ₄	+0.19		

4--In the same spirit, if the authors could provide Density of States (DoS) pictures out of DFT calculations it would certainly help the readers to better understand the electronic structure of those SACs.

Response: Thanks for your suggestion. DOS of some adsorbed structures (hydrogen, acetylene, and acetylene-hydrogen co-adsorption) has been provided in our original main text as rough estimations of electron excitation before our discussion about electron transfer using CDFT. According to the reviewer's suggestion, we have performed further calculation and provided all DOS into the revised Supplementary Fig.17- Fig. 26.

Figure R2. Projected density of states of pristine catalysts. Fermi levels are shifted to 0 eV. Due to there is only one Pd atom in models, to make PDOS of Pd can be clearly seen, PDOS of Pd is rescaled via normalization (PDOS/max(PDOS)) and multiplied by factor 0.02. Significant distribution of states projected on Pd on high energy level indicates high activity towards adsorption. For Pd/a101-1 and Pd/a101-2, bonding orbitals between Pd and support can be found in range -8 ~ -2 eV, corresponding anti-bonding orbitals can be seen in range 2~8 eV. For Pd/r110, additionally at about -0.5 eV, interaction between Pd and (Ti, O) can be seen.

Figure R3. Projected density of states of H₂ adsorbed structures. Fermi levels are shifted to 0 eV. Due to there are only two H atoms and one Pd atom in models, to make PDOS of H and Pd can be clearly seen, PDOS of H and Pd are rescaled via normalization (PDOS/max(PDOS)) and multiplied by factor 0.02. In range -8 ~ 8 eV, significant bonding between Pd and H can be seen in both Pd/a101-1 and Pd/r110 models. On the other hand, gaps between the highest occupied states projected onto Pd or Ti and the lowest unoccupied states of H are approximately larger than 8 eV, corresponding to a wavelength shorter than 155.30 nm.

Figure R4. Projected density of states of acetylene adsorbed structures. Fermi levels are shifted to 0 eV. Due to there are only two C atoms and one Pd atom in models, to make PDOS of C and Pd can be clearly seen, PDOS of C and Pd are rescaled via normalization (PDOS/max(PDOS)) and multiplied by factor 0.02. At about 0 eV and -3 eV, mixing of orbitals between Pd and C can be observed. States projected onto Pd in range -2 ~ 0 eV are corresponding well with those in Figure R2 in nearly the same range. However, gaps between the highest states and lowest unoccupied states projected onto C atoms are quite large, which implies impossibility of $C\equiv C$ bond activation due to electron excitation in this model.

Figure R5. Projected density of states of dissociated hydrogen and acetylene co-adsorbed structures. Fermi levels are shifted to 0 eV. Due to there are only two C atoms and one Pd atom in models, to make PDOS of C and Pd can be clearly seen, PDOS of C and Pd are rescaled via normalization ($\text{PDOS}/\max(\text{PDOS})$) and multiplied by factor 0.02. In all these three models, $\text{C}\equiv\text{C}$ activation due to electronic excitation is possible because gaps between the highest occupied states projected onto Ti or Pd (for example see Figure R8) and the lowest unoccupied states projected onto C (for example see Figure R9) are approximately 4 eV, corresponding to a wavelength about 310 nm.

Figure R6. Projected density of states of ethylene adsorbed structures. Fermi levels are shifted to 0 eV. Due to there are only two C atoms and one Pd atom in models, to make PDOS of C and Pd can be clearly seen, PDOS of C and Pd are rescaled via normalization (PDOS/max(PDOS)) and multiplied by factor 0.02. The lowest unoccupied states projected onto C has major contribution of p orbitals corresponding pi bond between C atoms. One can also find that low occupied states projected onto C atoms nearly retain property of isolated molecule, for example, states near -7 eV. However, gap between the highest occupied states projected onto Pd or Ti and pi* orbital at least 6 eV (corresponding wavelength 207 nm) approximately.

Figure R7. Projected density of states of dissociated hydrogen and ethylene co-adsorbed structures. Fermi levels are shifted to 0 eV. Due to there are only two C atoms and one Pd atom in models, to make PDOS of C and Pd can be clearly seen, PDOS of C and Pd are rescaled via normalization ($\text{PDOS}/\max(\text{PDOS})$) and multiplied by factor 0.02. Similar with Figure R5, gaps between the highest occupied states projected onto Pd or Ti (for example, see Figure R10) and the lowest unoccupied states projected onto C (for example, see Figure R11) are 3 eV approximately, corresponding a wavelength 414 nm.

Figure R8. The highest occupied state of model a101.Pd-2-ace-2H.

Figure R9. The lowest unoccupied states projected onto C atoms in model a101.Pd-2-ace-2H. A feature of π^* orbital symmetry can be seen clearly.

Figure R10. The highest occupied state of model a101.Pd-2-eth-2H.

Figure R11. The lowest unoccupied states projected onto C atoms in model a101.Pd-2-eth-2H.
A feature of π^* orbital symmetry can be seen clearly.

5--A more technical details on the DFT models. Do the authors have to apply for dipolar correction on the support cell to estimate properly the energy? It is not obvious that polarization effect upon Pd adsorption can be neglected.

Response: Thank you for your question. Yes, throughout our calculations of slab models, dipolar correction has been added.

6--There are several reports in the literature now, proving experimentally and theoretically that H₂ cannot dissociate easily on SAC, but rather on metal NPs. I think that fact deserves an additional comment.

Response: Thanks for your good suggestion. In many cases, the dissociation of H₂ usually goes to a heterolytic pathway on SACs, overcoming a higher barrier than that on the NPs with homolytic dissociation. Lacking cooperation from neighboring metal atoms and losing metallic property, may decrease the ability of single metal atoms to dissociate H₂, leading to a lower intrinsic activity for SACs. That's the reason why SACs often show higher selectivity but usually lower activity for semi-hydrogenation reaction (complete conversion at temperature above 120 °C) [*Chem. Rev.* 2020, 120, 683-733], and the reason why we would like to integrate the photo catalysis in this work to promote the activity of Pd SACs for semi-hydrogenation. We have added this discussion into our revised manuscript: However, the dissociation of H₂ usually goes to a heterolytic pathway on SACs, overcoming a higher barrier than that on the NPs with homolytic dissociation.[*Chem. Rev.* 2020, 120, 683-733] Lacking cooperation from neighboring metal atoms and losing metallic property, may decrease the ability of single metal atoms to dissociate H₂, leading to a lower intrinsic activity for SACs. Recently, photo-thermo catalysis, i.e., integrating photo- and thermo-catalysis to boost catalytic performance at more mild reaction condition, has attracted growing attention. [*Angew. Chem. Int. Ed.*, 2020, 59 (21), 8016-8035]

7--Thermodynamics aspects of the C₂H₂ and H₂ adsorption on Pd1 site are missing in the discussion. Could the authors provide such information?

Response: The reviewer raised a very good suggestion. It is well known that on Pd nanoparticle catalyst both ethyne and H₂ are adsorbed on Pd metal: Ethyne adsorbed by di-σ

bond which has a much higher intensity than H₂ adsorption (homolytic dissociation adsorption). Therefore, any changes of C₂H₂ or H₂ adsorption will have strong influence on each other and subsequently the catalytic activity. However, on Pd SACs, ethyne usually adsorbs by π -bond on isolated Pd atoms while H₂ adsorbs on Pd atoms and the adjacent support through heterolytic dissociation adsorption, resulting in a noncompetitive adsorption of ethyne and H₂ [*Natl. Sci. Rev.* 5, 653-672]. Therefore, we believe that on Pd SACs the thermodynamics aspects of the C₂H₂ and H₂ adsorption will have limited effect on each other but are more determined by the electronic properties of Pd single atoms as well as the interaction between Pd atoms and ethyne or H₂, respectively.

8--A minor comment on the SI: table X1 would be more valuable if errors were given instead.

Response: Thanks for your suggestion; a modified version of Table X1 has been provided in our revised Supplementary Information.

Table X1 (revised) Structural characters obtained from simulation and errors respect to experiments.

Structural characters	Optimized (anatase)/ Angstrom	Error respect to experiment/ Angstrom	Optimized (rutile)/ Angstrom	Error respect to experiment/ Angstrom
Ti-O bond 1	1.9817	-0.0588 ^{b)} /-0.0279 ^{c)}	1.9513	+0.0027 ^{d)} /-0.0128 ^{e)}
Ti-O bond 2	1.9349	-0.0609 ^{b)} /-0.0139 ^{c)}	1.9841	+0.0041 ^{d)} /-0.0203 ^{e)}
Cell parameter a	3.7901	-0.0601 ^{b)} /-0.0126 ^{c)}	4.5999	+0.0062 ^{d)} /-0.0533 ^{e)}
Cell parameter b	3.7901	-0.0601 ^{b)} /-0.0126 ^{c)}	4.5999	+0.0062 ^{d)} /-0.0533 ^{e)}
Cell parameter c	9.4894	+0.1194 ^{b)} /-0.2584 ^{e)}	2.9575	-0.0012/-0.0117 ^{e)}
	9.4805 ^{a)}	+0.1105 ^{b)} /-0.2673 ^{e)}		

a) *Data obtained from DFT + U (ramping), Mulliken method, where $U_{eff, Ti} = 4.0$ eV, $U_{ramping} = 0.5$ eV. A larger deviation from experimental value indicates accuracy of optimization task will not be better if DFT + U is used.*

b) *Data resource: 10.1524/zkri.1923.58.1.522, COD ID: 1010942*

- c) *Data resource: <https://materialsproject.org/materials/mp-390/>*
- d) *Data resource: 10.1107/S010876819100335X, COD ID: 9015662*
- e) *Data resource: <https://materialsproject.org/materials/mp-2657/>*

Response to Reviewer 2

In this article, Li, Qiao et al report important results on the selective hydrogenation of acetylene, an important reaction in industry. The authors are based on convincing experimental results, supported by theoretical calculations to demonstrate a photo-thermal effect operating on ultradisperse Pd/TiO₂ catalysts. The significance of these results is in line with a publication in *Nat. Commun.* once the following changes have been introduced.

1) The proposed green chemistry approach should be discussed in the perspective of the following article: Room-temperature electrochemical acetylene reduction to ethylene with high conversion and selectivity.

Response: We truly thank the reviewer for this good suggestion. We have added the discussion in our revised manuscript: Diverse methods have been developed to eliminate the acetylene impurity among which the electrocatalysis has been proven a green chemistry approach. [*Nat. Catal.*, 2021, 4, 565-574; *Nat. Commun.*, 12, 7072. 10] For example, Zhang et al reported a room-temperature electrochemical reduction strategy of acetylene over a layered double hydroxide (LDH)-derived Cu catalyst, which manifests high catalytic performance but suffers from unaddressed issues for large-scale applications, such as the low cell energy efficiency [*Nat. Catal.*, 2021, 4, 565-574]. On the other hand, thermocatalytic semi-hydrogenation of acetylene into ethylene seems more efficient, and has been extensively applied in industry for decades.

2) The authors use as support a rutile/anatase mixture, which is certainly not innocent (cf. *J. Phys. Chem. B* 2006, 110, 15, 8019-8024; *J. Chem. Phys.* 136, 104107 (2012)). It is not clear in the manuscript whether the Pd SAs are localized on the rutile and/or anatase phase. Please use the EELS analyses to try to see more clearly (cf 10.1557/proc-139-327). Please also comment on the binding energies of the Pd SAs on these two phases. In order to be able to conclude on the nature of the active phase Pd SA/TiO₂-a or PdSA/TiO₂-r.

Response: This is a very good question. The XRD measurements have shown that our TiO₂ support after high-temperature reduction consists of both anatase and rutile. The AC-STEM image analysis also confirmed the mixture of both phases, Figure R12. Therefore, according to the literature reports and our study experience on TiO₂ supported metal single atoms, we

believe that Pd single atoms can locate at both anatase and rutile phase.

The reference provided by the reviewer [*J. Phys. Chem. B* 2006, 110, 15, 8019-8024] indeed suggests a TiO₂-phase-dependent selectivity in semi-hydrogenation of acetylene. However, it should be noted that the catalysts used in that work are quite different from ours. They used TiO₂ supported Pd nanoparticles on which both acetylene and ethylene are adsorbed preferentially through ethylidyne mode on 3-fold Pd sites and di- σ bond on bridged Pd dimers (Figure R13a and R13b). These types of adsorption are strong, resulting in a low ethylene selectivity that varies from largely negative value to positive value in the presence of excess ethylene due to the difficulty of ethylene desorption. Therefore, any change of either geometric or electronic structure of Pd, or any possible contribution from support may have large influence on the selectivity. On the contrary, the catalysts in our work are SACs where acetylene and ethylene are adsorbed mainly by π -bond, Figure R13c, leading to the easy desorption of ethylene and thus higher ethylene selectivity. Therefore, the better selectivity mainly comes from the special adsorption manner of ethylene on single atoms thus the contribution from supports might be faint. In fact, all the SACs reported so far have shown very good ethylene selectivity despite the supports used [*J. Am. Chem. Soc.* 2015, 137, 10484-10487; *Chin. J. Catal.* 2016, 37, 692-699], further verifying our conjecture.

To further convince the reviewer we have performed additional control experiment. It is generally accepted that anatase (or mixture of anatase and rutile with suitable ratio, such as P25) is more suitable as support to load metal atoms than rutile. Therefore, although it's hard to obtain pure anatase supported catalyst after high-temperature treatment due to the phase transformation from anatase to rutile, we can prepare rutile supported Pd catalyst. If rutile supported Pd SAC shows similar performance, it means that support doesn't have significant influence. We therefore prepared a rutile supported Pd catalyst and reduced at different temperatures and tested their performance. The results show a similar scenario that the ethylene selectivity after reduction at a higher temperature 700 °C is much improved compared with that reduced at 200 °C, Figure R14. A slightly higher reduction temperature of 700 °C was adopted is because rutile is slightly harder to form SMSI than anatase and P25 [*Sci. Adv.*, 2017, 3(10): e1700231; *Angew. Chem. Int. Ed.*, 59, 11824-11829]. We have added the discussion in revised Supplementary file (Supplementary Fig. 9).

Figure R12. AC-STEM images of Pd/TiO₂-600H.

Figure R13. Adsorption patterns of ethylene on Pd catalysts with different geometric structures.

Figure R14. Acetylene conversion as a function of temperature for acetylene semi-hydrogenation (a), and ethylene selectivity (b) over 0.15 wt% Pd/TiO₂(rutile)-700H and 0.15 wt% Pd/TiO₂(rutile)-200H. Reaction conditions: 1 vol% C₂H₂, 10 vol% H₂, 20 vol% C₂H₄ balanced with He; WHSV = 180 000 mL·h⁻¹·gcat⁻¹

Response to Reviewer 3

In this work, Guo et al. demonstrated semi-hydrogenation of acetylene in excess ethylene on atomically dispersed Pd species on TiO₂ supports prepared by two different methods. The authors exploited the strong metal-support interaction (SMSI) to selectively encapsulate Pd nanoparticles and clusters by the SMSI overlayer via high-temperature reduction of the catalysts in H₂, leaving single-atom Pd species exposed for catalysis. This treatment resulted in substantial improvement in C₂H₄ product selectivity in thermocatalysis. The selectivity gradually dropped from 60% to about 40% over 40 hours of reactivity test with stable %100 C₂H₂ conversion. Next, the authors demonstrated that high energy near UV photons could be utilized to drive the same reaction at milder operating conditions. Supported by their computational and experimental studies, the authors suggest that the origin of the reactivity enhancement in photocatalysis is the photoexcitation of TiO₂ and the transfer of charge carriers to Pd sites for chemical bond activation (primary to activate C-C bonds). The combination of the SMSI and light to control catalysis is interesting! However, there are significant concerns on the photocatalysis side and the associated discussion. In particular, an important aspect of the photocatalysis results that the authors seemingly refused to acknowledge is regarding the selectivity of the process. Light can promote the reactivity but also lead to a substantial drop in desired product selectivity in this case (see below). This poses a major concern for using the current photocatalysis approach for driving the semi-hydrogenation of acetylene on the proposed catalyst system.

1. Comparison of the results in Figure 4 and S8 (e.g., focusing on their panel (a)) reveals that light-promoted reactivity comes at the expense of a substantial drop in the C₂H₄ product selectivity. Looking at the entire 35 min of the experiment, the selectivity decreases from 80% to near 40%. Even with the assumption that some drop in selectivity during the initial light-on cycle is due to measurement under non-steady-state condition (that has its own issue), it is clear that there is an additional 10% drop in the selectivity each time that light is turned on with a total of near 20% selectivity drop just in two illumination cycles. Regrettably, the authors consider this as a “slight” decrease in selectivity or even “stable” selectivity (during the second cycle)!

Response: We sincerely thank the reviewer for the good questions and suggestions mainly

regarding to ethylene selectivity.

In fact, the semi-hydrogenation of ethyne in excess ethylene over Pd metal can be regarded as a “preferential” hydrogenation process, similar to the preferential oxidation of CO in excess of H₂. The reason we must remove ethyne from excess ethylene (or CO from excess H₂) is their adsorption (ethyne or CO) on metal is much stronger than their counterparts (ethylene or H₂), thus preferentially covering the catalyst active sites and inhibiting the conversion of alkene (H₂), giving rise to the so-called catalyst poison. Therefore, the conversion of alkene (reflected as alkene selectivity) is highly alkyne-coverage-dependent. With the acetylene conversion increase (coverage decrease), the possibility of ethylene adsorption and activation increase, i.e., increased acetylene conversion will result in a decreased ethylene selectivity (especially at relatively higher conversion). This is a common phenomenon in ethyne semi-hydrogenation reaction in rich ethylene atmosphere [*Catal. Rev.* 2006, 48, 91-144; *Frontiers of Chemical Science and Engineering* 2015, 9, 142-153]. Therefore, the selectivity drop might mainly originate from the conversion increase upon photo-thermo condition. At full ethyne conversion, ethylene can adsorb and undergo hydrogenation, leading to a sharp decrease of ethylene selectivity, that’s the reason why on Pd NPs the ethylene selectivity can quickly drop from >90% to as low as -500% in our work (Figure 3b). Theoretically, in our test condition the ethylene selectivity could vary from -800% to 100%, a very large range. Therefore, 20% change is small compared with such a large range ($20/900 \times 100\% = 2.2\%$). That’s the reason we claim a small selectivity change. But according to the reviewer’s comments, we have deleted this word to avoid any possible misleading.

As to the claim of “stable” in second cycle, it is based on the experimental data that in first cycle (either light off or light on) both activity and selectivity changed gradually (activity increase while selectivity decrease), suggesting a change of the catalyst under reaction condition. However, in second cycle both activity and selectivity are stable despite the different values in light on and light off conditions, suggesting the catalyst became stable after first cycle. To further convince the reviewer, we have conducted another photo-thermo catalytic experiment with more cycles. As shown Figure R15, the catalyst is quite stable after first cycle. We have added the new data and discussion into our revised Supplementary file

(Supplementary Fig. 12).

Figure R15. Catalytic Performance of photo-thermo catalysis over Pd/TiO₂-600H with more cycles.

2. The drop in selectivity also implies that additional light-on cycles (if performed) could potentially lead to a further decrease in selectivity.

Response: As mentioned above, we think the selectivity change mainly comes from the ethyne conversion increase. Therefore, in additional light-on cycles the selectivity should not change much if conversion didn't change much. This can be confirmed by the additional cycle experiment results, Figure R15.

3. Furthermore, the light-induced drop in the C₂H₄ product selectivity is an important result that is deserved to be highlighted in the main text along with the reactivity results in photocatalysis.

Response: Thank you for your suggestion, we have added corresponding discussion in the revised manuscript: It turns out that the introduction of light irradiation can indeed boost the conversion of acetylene in semi-hydrogenation reaction remarkably (from 20% to 80%) with a decreased ethylene selectivity, Fig. 4 and Supplementary Fig. 10, indicating that the photogenerated charges may involve in the reaction process. The decreased selectivity upon irradiation might stem from the increased acetylene conversion as the ethylene selectivity is usually acetylene-conversion-dependent. [*Catal. Rev.* 2006, 48, 91-144]

4. Notably, the authors repeatedly state that there is only one report on heterogeneous

photocatalytic semi-hydrogenation of acetylene (ref. 26, Zhou et al., Adv Mater 2019) while oversight the previous original work in this area (Swearer et al., PNAS 2016) using visible light and a plasmonic photocatalyst for semi-hydrogenation of acetylene. Interestingly, unlike the current results, the work by Swearer et al. demonstrated that light could promote both the reactivity and the selectivity for this reaction.

Response: We sincerely thank the reviewer for pointing out this work that we had left out. We have cited this paper in our revised manuscript in introduction section. Swearer et al. [PNAS 2016, 113, 8916-8920] used Pd nanoparticles and aluminum nanocrystals (AINC) to construct a heterometallic antenna-reactor complexes photocatalyst for semi-hydrogenation of acetylene under white-light illumination by controlling the surface plasmon resonance. They found ethylene selectivity shows an increase with increased laser power density but a limited yield differing from traditional thermal catalysis under the same reaction conditions. The reason is mainly that plasmon-induced hot carriers improve the desorption efficiency of H₂, limiting the additional hydrogenation of surface ethylene, but also limiting the hydrogenation of acetylene, leading to a low target product yield simultaneously. Their study is based on two preconditions, ethylene desorption and hydrogenation on Pd NPs have similar activation barriers, and the small change of dissociated H₂ has big influence on ethylene desorption or over-hydrogenation. But in our work, the catalyst is totally different. As can be seen in our answer to question 7 of reviewer #1, ethyne adsorbed by di-σ bond which has a much higher intensity than H₂ adsorption (homolytic dissociation adsorption). Therefore, any changes of C₂H₂ or H₂ adsorption will have strong influence on each other and subsequently the catalytic activity. However, on Pd SACs, ethyne usually adsorbs by π-bond on isolated Pd atoms while H₂ adsorbs on Pd atoms and the adjacent support through heterolytic dissociation adsorption, resulting in a noncompetitive adsorption of ethyne and H₂ on SACs [Natl. Sci. Rev. 5, 653-672]. Therefore, minimal change of dissociated H₂, if any, will not determine the ethylene tendency towards desorption or over-hydrogenation. Besides, our H₂-D₂ isotope exchange reaction had demonstrated that H₂ dissociation ability is almost the same upon light irradiation or not. However, in an ethylene-rich condition, with the increasing of acetylene conversion, the possibility of ethylene to adsorb and activate on active sites is increasing, i.e., increased acetylene conversion will result in decreased ethylene selectivity.

5. The authors should also discuss how the use of light in this system leads to worsened C₂H₄ selectivity. Reportedly, the results of DFT calculations and H₂-D₂ exchange experiment suggest the primary role of Pd/TiO₂ photoexcitation being the C-C bond activation acetylene activation, but the discussion of product selectivity is missing.

Response: Thank you for this question which is actually very similar or closely related to your question 1 and 2. As can be seen in answers to question 1 and 2, the worsened C₂H₄ selectivity comes from the increased ethyne conversion as in additional light-on cycles ethylene selectivity didn't decrease further.

6. Regarding the authors' discussion on the role of light and the underlying mechanism: It is stated that the reactivity enhancement is due to a charge-driven mechanism from photoexcitation of TiO₂ and as supported by the authors XPS and DFT results. This statement is broadly true. However, there are important aspects of light-induced catalyst heating that should be considered when discussing the role of light in photocatalytic processes. For instance, despite the authors' statement, the use of a circulating water bath does not necessarily exclude the photothermal heating effect in catalysis. This is due to the potential presence of local nanoscale hot spots, especially with supported metal catalysts. Also, the magnitude of the light-induced heating within the catalyst bed strongly depends on multiple factors, including the optical properties (ensemble photon absorption) and thermal conductivity of the catalyst, incident photon flux, illumination wavelength, etc. Currently, the information regarding the power density of the incident photons in Figure 4 is missing. Also, the use of Al₂O₃ support in the control experiment and the lack of photocatalytic reactivity does not necessarily exclude the thermal effect on TiO₂ since this is a comparison of two different systems with a very different absorption properties and thermal conductivities.

Response: We sincerely thank the reviewer for this critical comment and good suggestion. The light-induced heating effect is one of the branches in photo-thermo catalysis. [*Adv Mater*, 2019, e1900509] That's the reason we had tried to verify or exclude the potential heat effect in our original manuscript by performing various control experiments. The results together with the following reasons make us believe that the heat effect might not contribute

significantly.

First, the light-to-heat conversion efficiency of normal TiO₂ is usually not high. For example, Branislav et al. [*Applied Materials Today* 2020, 20, 100669] reported a quick temperature increase from 25.4 °C to 90 °C on black TiO₂ (BTiO₂) film with random size distribution under 1 sun illumination. However, for the corresponding white crystalline TiO₂ (WTiO₂) with solely anatase phase, the temperature increase is only 15 °C (from 25 to 40 °C). As we used white TiO₂ crystal as support, we believe that the heat effect will not significant. Second and perhaps of more importance, to avoid any possible heat effect, we have used a differential reactor with a very low catalyst bed (<1 mm) and have also highly diluted the catalyst bed by using dozens of times of diluter (for example, 10 mg catalyst diluted by 400 mg quartz sand). In this case, we think the heat effect, if any, could be minimized.

According to reviewer's suggestion, we have added this discussion into our revised manuscript. We have also supplied the power density information for Fig. 4 and Supplementary Fig. 10 in the revised manuscript.

7. Furthermore, the authors state that "it is suggested that the light can still increase the reactor temperature by 10 °C below 100 °C. Therefore, a 10 °C-correction was used when performing photo-thermo catalysis." Suggested by who and based on what evidence? Again, the magnitude of light-induced heating depends on several factors, as discussed above. Also, did the authors apply this "undocumented" correction to the Arrhenius plot of photocatalysis in Figure 4e?

Response: Thank you for your good questions. We are sorry for not making this clearly in our original manuscript. A more detailed picture of our fixed-bed reactor has been provided in the revised Supplementary Fig. 11. As mentioned above, the 10 °C-correction below 100 °C was obtained based on our oft-repeated experiments. As shown in Figure R16, thermocouple-1 was put inside of the furnace for temperature setting while thermocouple-2 was put into the reactor tube just upon the catalyst thin layer to measure the temperature of catalyst bed. In the case without introducing light, both thermocouples show almost same temperature. However, when introducing light irradiation, thermocouple-2 usually shows a 10 °C-increase compared with thermocouple-1 at temperature below 100 °C, regardless of the catalysts used. This

temperature increase might come from the visible and near-infrared light radiation as the light was only introduced into the inside of the reactor rather than the outside. Such a slight temperature increase has been reported frequently in previous literatures. [Joule, 2019, 3(4):920-937, *Angew. Chem. Int. Ed.* 2020, 59, 12909-12916] Thus a 10 °C-correction is always used in our photo-thermo catalysis reaction, i.e., controlling the catalyst bed temperatures same with or without introducing light irradiation by adjusting furnace temperature (10 °C lower), to exclude the overall light-induced heating effect which was also used in previous reports. [*Angew. Chem. Int. Ed.* 2020, 59, 12909-12916] And of course, this correction had been used to the Arrhenius plot.

Figure R16. The fixed-bed reactor (a) and amplification of the furnace part (b) for photo-thermo semi-hydrogenation of acetylene.

8. The authors suggested a charge carrier-driven mechanism for bond activation under illumination with charge transfer from TiO₂ to Pd sites. They should also provide a scheme to discuss the proposed mechanism further, especially to examine whether there is a net charge transfer process to adsorbate(s) and that the catalyst remains in a neutral state or not. If the process is associated with net charge transfer, it results in a charge build-up and eventually a decrease in photocatalytic reactivity. A long-term stability test in photocatalysis should

therefore be provided.

Response: Thanks for your good suggestion. The possible mechanism for photo-thermo catalysis over the constructed Pd₁/TiO₂ is proposed in Figure R17 (Supplementary Fig. 30 in the revised version). The scheme can be described as follows.

1) Irradiation leads to band-gap excitation of TiO₂ support, generating electron-hole pairs: $\text{TiO}_2 + h\nu \rightarrow \text{TiO}_2 (e^-, h^+)$

2) Photo-generated electrons transfer to isolated Pd sites, facilitating the activation of adsorbed C₂H₂: $\text{Pd} + e^- + \text{C}_2\text{H}_2 \rightarrow \{\text{Pd}-\text{C}_2\text{H}_2^-\}$

3) H₂ dissociation goes to a heterolytic pathway on isolated Pd sites and TiO₂ support and then photo-generated holes trap with the dissociated Pd-H⁻ to form Pd-H:

4) TiO₂-H⁺ and Pd-H react with the activated Pd-C₂H₂⁻ to form Pd-C₂H₄ species, followed by C₂H₄ desorption: $\{\text{Pd}-\text{C}_2\text{H}_2^-\} + \{\text{Pd}-\text{H}, \text{TiO}_2-\text{H}^+\} \rightarrow \text{Pd}-\text{C}_2\text{H}_4$

According to this mechanism, every electron transfer is accompanied with a hole and the net charge is noncumulative. Therefore, our understanding is that upon reaction condition, the net charge is noncumulative. However, without the presence of reactants, a certain degree of net charge accumulation is not impossible as the Pd atoms accepted some electrons from the support, resulting in a difficulty in electron-hole recombination. To further confirm the result, we tested the stability of a Pd/TiO₂-H200 according to the reviewer's suggestion (to avoid any potential effect of light on the SMSI state and then on the catalytic performance). It can be seen that the catalytic performance is quite stable upon light irradiation after an evaluation in the initial few hours in dark condition, as shown in Figure R18. We have added this data into the revised Supplementary file (Supplementary Fig. 31).

Figure R17. The possible scheme of photo-thermo acetylene semi-hydrogenation over Pd₁/TiO₂ SAC.

Figure R18. Acetylene conversion and ethylene selectivity over the catalyst of 0.1 wt% Pd/TiO₂ NP-200H at 65 °C in the dark and upon a full-spectrum light irradiation, power density: 167 mW · cm⁻², WHSV = 720 000 mL · h⁻¹ · g_{cat}⁻¹. Reaction conditions: 1 vol% C₂H₂, 10 vol% H₂, 20 vol% C₂H₄ balanced with He.

9. Spectrum of the light source used for illumination is missing.

Response: Thanks for reminding us about the missing spectrum, we have added the following spectrum (Figure R19) to Supplementary Fig. 29 in the revised version.

Figure R19. Spectrum of the light source used for illumination.

10. Intensity-dependent photocatalytic studies should provide more insights regarding light-matter interactions for photocatalysis in this system?

Response: Thanks for your suggestion, we have added the intensity-dependent experiment (Figure R20) to Supplementary Fig. 13 in the revised version.

Figure R20. Intensity-dependent experiment upon a full-spectrum light irradiation. **a** Acetylene conversion and **b** ethylene selectivity as a function of power density over the catalyst of 0.15 wt% Pd/TiO₂-600H at 70 °C (the test temperature). Reaction conditions: 1 vol% C₂H₂, 10 vol% H₂, 20 vol% C₂H₄ balanced with He; WHSV = 360 000 mL·h⁻¹·g_{cat}⁻¹.

11. Improper citations! The past decade has witnessed huge progress in light-driven heterogeneous catalysis, especially with supported metal catalysts, using a combination of light and external heating or only light to drive the chemistry (the latter is more ideal). The authors must more appropriately acknowledge the previous efforts in this area rather than citing a few very recent examples as the pioneering work in the field, which is indeed an incorrect remark!!!

Response: Thank you for pointing out our improper citations, we have cited more previous works of light-driven heterogeneous catalysis in our revised manuscript: photogenerated carriers can directly transfer into the orbitals of adsorbed molecules to promote their desorption, dissociation, or activation thus trigger the chemical reaction, giving rise to a totally different reaction way. [*Nat. Energy.*, 2020, 5 (1), 61-70; *Nano Lett.* 2016, 16 (10), 6677-6682; *Science*, 2018, 362 (6410), 69-72; *Adv Sci*, 2014, 1 (1), 1400001] Recent pioneering studies have demonstrated that the coupling of thermo- and photocatalytic processes overcomes the low activity in photocatalysis and high reaction barrier in thermocatalysis, thus offering a promising strategy to promote the activity and/or selectivity for various meaningful reactions, such as hydrogenation, oxidation, CO₂ reduction, Fischer-Tropsch synthesis, water-gas shift (WGS) reaction. [*Angew. Chem. Int. Ed.*, 2020, 59 (21), 8016-8035; *Energy Environ. Sci.*, 2019, 12 (4), 1122-1142; *Chem. Soc. Rev.*, 2021, 50, 2173-2210; *Angew. Chem. Int. Ed.*, 2019, 58 (23), 7708-7712; *Catal. Chem.*, 2018, 4 (12), 2917-2928; *Nat. Commun.*, 2017, 8, 14542; *Science*, 2013, 339 (6127), 1590-3; *ACS Catal*, 2017, 7, 7461-7465.]

12. The authors consider photogenerated carriers in TiO₂ as “hot carriers”. This should not be confused from well-known hot carrier generation in plasmonic metal nanostructure.

Response: Thank you for pointing out this, we have changed “hot carriers” to “photogenerated carriers” in our revised manuscript to avoid the confusion.

13. Words such as “obvious”, “clearly”, etc. are overused throughout the manuscript. Their use should be minimized as they do not necessarily always reinforce the best scientific scenario.

Response: Thanks for the reviewer’s good suggestion. We have reduced the repeated use of adjectives and adverbs such as “obvious”, “clearly”, “mainly” in our revised manuscript.

REVIEWERS' COMMENTS

Reviewer #1 (Remarks to the Author):

Since all the issues addressed by the different referees have been considered by the authors and since they have largely improved the manuscript in consequences, I'm pleased to recommend the present form of the manuscript for publication in Nat. Commun.

Reviewer #2 (Remarks to the Author):

The authors have answered the questions quite convincingly, so this article can be published. I would nevertheless suggest including in the ms the proposed scheme of reaction mechanism (Fig S30) after touching it up a bit to make it clearer.

Reviewer #3 (Remarks to the Author):

The authors response to my concerns and comments is adequately satisfactory. I recommend the revised manuscript for publication.

RESPONSE TO REVIEWERS' COMMENTS

We appreciate all the reviewers for their positive comments and the constructive suggestion which are helpful to report our manuscript. A point-by-point response to the reviewers' comments is as following:

Reviewer #1 (Remarks to the Author):

Since all the issues addressed by the different referees have been considered by the authors and since they have largely improved the manuscript in consequences, I'm pleased to recommend the present form of the manuscript for publication in Nat. Commun.

Response: We sincerely thank the reviewer for this encouraging comment, and appreciate your recommendation.

Reviewer #2 (Remarks to the Author):

The authors have answered the questions quite convincingly, so this article can be published.

I would nevertheless suggest including in the ms the proposed scheme of reaction mechanism (Fig S30) after touching it up a bit to make it clearer.

Response: We truly thank the reviewer for this good suggestion and encouraging comment. We have touched the proposed scheme of reaction mechanism up a bit (shown in Fig. RX1). The manuscript is revised in P18 as following:

A possible mechanism as well as process is proposed in Supplementary Fig. 30, described as following: 1) Irradiation leads to band-gap excitation of TiO₂ support, generating electron-hole pairs; 2) H₂ dissociation goes to a heterolytic pathway on isolated Pd sites and TiO₂ (at the Pd-O interface); 3) C₂H₂ molecules adsorb on isolated Pd sites and then photo-generated electrons transfer to isolated Pd sites, facilitating the activation of adsorbed C₂H₂; 4) H species react with the activated C₂H₂ on Pd single atoms to form Pd-C₂H₄ species, followed by C₂H₄ desorption. According to this mechanism, there should not be net charge accumulation.

We hope the modified one can meet your expectation.

Fig. RX1 The detailed mechanism and process of photo-thermo acetylene semi-hydrogenation over Pd₁/TiO₂ SAC

Reviewer #3 (Remarks to the Author):

The authors response to my concerns and comments is adequately satisfactory. I recommend the revised manuscript for publication.

Response: We truly thank the reviewer for this positive comment, and appreciate your recommendation.